# VAP–SCRN1 interaction regulates dynamic endoplasmic reticulum remodeling and presynaptic function

Feline W Lindhout[1] (iD), Yujie Cao[1,†], Josta T Kevenaar[1,†], Anna Bodzęta[1,†], Riccardo Stucchi[1,2], Maria M Boumpoutsari[1] (iD), Eugene A Katrukha[1], Maarten Altelaar[2] (iD), Harold D MacGillavry[1] & Casper C Hoogenraad[1,3,*] (iD)

## Abstract

In neurons, the continuous and dynamic endoplasmic reticulum (ER) network extends throughout the axon, and its dysfunction causes various axonopathies. However, it remains largely unknown how ER integrity and remodeling modulate presynaptic function in mammalian neurons. Here, we demonstrated that ER membrane receptors VAPA and VAPB are involved in modulating the synaptic vesicle (SV) cycle. VAP interacts with secernin-1 (SCRN1) at the ER membrane via a single FFAT-like motif. Similar to VAP, loss of SCRN1 or SCRN1-VAP interactions resulted in impaired SV cycling. Consistently, SCRN1 or VAP depletion was accompanied by decreased action potential-evoked $Ca^{2+}$ responses. Additionally, we found that VAP–SCRN1 interactions play an important role in maintaining ER continuity and dynamics, as well as presynaptic $Ca^{2+}$ homeostasis. Based on these findings, we propose a model where the ER-localized VAP–SCRN1 interactions provide a novel control mechanism to tune ER remodeling and thereby modulate $Ca^{2+}$ dynamics and SV cycling at presynaptic sites. These data provide new insights into the molecular mechanisms controlling ER structure and dynamics, and highlight the relevance of ER function for SV cycling.

**Keywords** axon; endoplasmic reticulum; secernin; synaptic vesicle cycle; VAP
**Subject Categories** Membranes & Trafficking; Neuroscience
**The EMBO Journal (2019) 38: e101345**

## Introduction

The continuous and dynamic ER network is one of the most abundant organelles in cells. In neurons, somatodendritic domains contain both rough and smooth ER, whereas axons exclusively exhibit smooth ER. The smooth ER lacks ribosomes and is not involved in translation; instead, it is important for $Ca^{2+}$ homeostasis, lipid synthesis and delivery, and signaling. The relevance of axonal ER in particular is highlighted by various human axonopathies caused by mutations in different generic ER proteins. More specifically, dysfunction of ER-shaping proteins such as atlastin-1, reticulon-2, receptor expression-enhancing protein 1 (REEP1), and receptor expression-enhancing protein 2 (REEP2) leads to hereditary spastic paraplegia (HSP), whereas mutations in ER receptor VAMP-associated protein B (VAPB) cause amyotrophic lateral sclerosis (ALS; Hazan *et al*, 1999; Zhao *et al*, 2001; Nishimura *et al*, 2004; Zuchner *et al*, 2006; Montenegro *et al*, 2012; Esteves *et al*, 2014; Yalcin *et al*, 2017). Together, these pathologies hint for an increased sensitivity for proper ER structure and function in axons.

Recent ultrastructural three-dimensional analysis revealed that the ER in axons is comprised of a conserved and unique organization (Wu *et al*, 2017; Yalcin *et al*, 2017; Terasaki, 2018). The axonal ER structure consists of narrow ER tubules, which occasionally form cisternae at tubular branch points with comparably small lumen (Wu *et al*, 2017; Yalcin *et al*, 2017; Terasaki, 2018). This distinctive ER network extends throughout all axon branches with a relative constant density of only 1–2 narrow tubules per diameter, while remaining continuous with the rest of the ER network (Wu *et al*, 2017; Yalcin *et al*, 2017; Terasaki, 2018). At presynaptic terminals, the ER forms a local tubular network opposing the active zone. This presynaptic ER structure often wraps around mitochondria and is in close proximity to the plasma membrane, and it regularly forms tight membrane contact sites with these structures (Wu *et al*, 2017; Yalcin *et al*, 2017). Moreover, fast dynamics of axonal ER was observed in *Drosophila* neurons using fluorescent recovery after photo-bleaching (FRAP) analysis, suggesting that the neuronal ER network likely undergoes dynamic remodeling (Wang *et al*, 2016; Yalcin *et al*, 2017). However, the precise role of the dynamic ER

1  Department of Biology, Cell Biology, Utrecht University, Utrecht, The Netherlands
2  Biomolecular Mass Spectrometry and Proteomics, Bijvoet Center for Biomolecular Research and Utrecht Institute for Pharmaceutical Sciences, Utrecht University, Utrecht, The Netherlands
3  Department of Neuroscience, Genentech, Inc., South San Francisco, CA, USA
   *Corresponding author. Tel: +31 030 2533894; Fax: +31 030 2513655; E-mail: c.hoogenraad@uu.nl
   †These authors contributed equally to this work

network in axons and at presynaptic sites remains poorly understood.

Emerging evidence implies that the presynaptic ER is engaged in modulating the tightly controlled $Ca^{2+}$-induced SV cycle (Summerville *et al*, 2016; De Gregorio *et al*, 2017; de Juan-Sanz *et al*, 2017). In *Drosophila* neurons, it was reported that homologues of the HSP-associated ER-shaping proteins atlastin-1 and reticulon-1 are implicated in controlling neurotransmitter release at neuromuscular junctions, as loss of these proteins resulted in a marked decrease in SV cycling (Summerville *et al*, 2016; De Gregorio *et al*, 2017). In mammalian neurons, recent reports showed that presynaptic $Ca^{2+}$ levels in the ER are locally elevated during evoked neuronal transmission, suggesting that the presynaptic ER buffers $Ca^{2+}$ to modulate SV cycling (de Juan-Sanz *et al*, 2017).

Moreover, the ER transmembrane protein VAP was originally identified as regulator of synaptic transmission in *Aplysia californica*, where it was specifically expressed in neuronal tissue (Skehel *et al*, 1995). Conversely, mammalian VAPA and VAPB are ubiquitously expressed in different cell types and its intracellular localization is restricted to ER membranes. VAPs act as key players in facilitating tight membrane contact sites between the ER and other intracellular membranes, which represent functional interactions through which $Ca^{2+}$ exchange and lipid transfer occur (Muallem *et al*, 2017; Wu *et al*, 2018). VAP contains a C-terminal transmembrane domain which is inserted into the ER membrane, and a cytoplasmic N-terminal tail with a coiled-coil domain and a major sperm protein (MSP) domain. The MSP domain exhibits a FFAT(-like) binding site, which is unique for VAP proteins. Many VAP-associated proteins (> 100) with such a FFAT(-like) motif have been described (Murphy & Levine, 2016). This includes the cytoplasmic protein SCRN1, which contains a N-terminal C69 domain and a C-terminal coiled-coil domain and was predicted to have FFAT(-like) motifs (Murphy & Levine, 2016). The large number of FFAT-containing proteins typically localize to distinct subcellular

structures, which has led to the general idea that VAP may act as a key ER receptor.

In this study, we demonstrated that ER membrane protein VAP and cytoplasmic VAP-associated protein SCRN1 are important for $Ca^{2+}$-driven SV cycling. We found that VAP interacts with SCRN1 at the ER membrane through a single FFAT-like motif. Decreasing these ER-localized VAP–SCRN1 interactions was accompanied by a number of phenotypes, including discontinuous ER tubules, impaired ER dynamics, elevated basal presynaptic $Ca^{2+}$ levels, and decreased SV cycling. Together, these data point toward a model where ER remodeling, mediated by VAP–SCRN1 interactions is engaged in modulating $Ca^{2+}$ dynamics and SV cycling at presynaptic sites.

## Results

### ER proteins VAPA and VAPB are involved in regulating SV cycling

To determine whether the ER proteins VAPA and VAPB could be involved in modulating presynaptic function, we first mapped their subcellular localization in primary rat hippocampal neurons. Similar as reported previously in *Drosophila* neurons, we found that endogenous VAPA and VAPB appeared as punctae present along ER structures in axons which often co-localized with presynaptic marker synaptotagmin (Syt; Pennetta *et al*, 2002). At somatodendritic regions, endogenous VAPA and VAPB revealed a more diffuse patchy staining pattern that co-localized with expressed ER membrane protein Sec61β (Fig 1A). Exogenous HA-VAPA and HA-VAPB were observed at ER structures throughout neurons and also partially co-localized with presynaptic boutons (Fig EV1A). Thus, VAP is abundantly present at ER structures throughout the cell including at presynaptic sites. To test whether VAP could be engaged in regulating synaptic functions, we next investigated

▶

**Figure 1. VAP and VAP-associated protein SCRN1 modulate SV cycling.**

A   Endogenous localization of VAPA or VAPB and Syt in hippocampal neurons (DIV16) expressing GFP-Sec61β. Zooms represent (1) an axonal structure with presynaptic boutons (arrowheads), and (2) a dendritic structure. Scale bars: 10 μm (full size) and 5 μm (zoom).
B   Schematic illustration of the Syt antibody uptake assay: live neurons were stimulated with bicuculline and incubated with primary Syt antibodies, and next neurons were fixed and stained with secondary antibodies.
C   Representative image of Syt antibody uptake at axons of hippocampal neurons (DIV18) co-expressing RFP and pSuper empty vector or VAPA/B shRNAs. Yellow and gray arrowheads mark presynaptic boutons with and without internalized Syt, respectively. Zooms represent typical boutons. Scale bars: 5 μm (full size) and 2 μm (zoom).
D   Quantifications of fluorescence intensity of internalized endogenous Syt at single presynaptic boutons of hippocampal neurons (DIV18) co-expressing RFP and pSuper empty vector or VAPA/B shRNAs. *N* = 2, *n* = 288–541 boutons.
E   Western blot of endogenous SCRN1 expression in indicated adult rat neuronal and non-neuronal tissues. Cereb., cerebellum. Hippoc., hippocampus. Spin., spinal.
F   Pull-down assay of HEK293T cells co-expressing Myc-VAPA with BioGFP or BioGFP-SCRN1.
G   Pull-down assay of HEK293T cells co-expressing Myc-VAPB with BioGFP or BioGFP-SCRN1.
H   Scaled representation of SCRN1-associated proteins identified with pull-down assay followed by mass spectrometry analysis of purified BioGFP or BioGFP-SCRN1 from HEK293T cell lysates with adult rat brain extracts. All candidates showed > 10 enrichment of PSM compared to control.
I   Endogenous localization of SCRN1 and Syt in cortical neurons (DIV18) expressing GFP. Zoom represents an axon structure with presynaptic boutons (arrowheads). Scale bars: 10 μm (full size) and 5 μm (zoom).
J   Representative image of Syt antibody uptake at axons of hippocampal neurons (DIV18) co-expressing RFP and pSuper empty vector, SCRN1 shRNA, or SCRN1 shRNA with GFP-SCRN1. Yellow and gray arrowheads mark presynaptic boutons with and without internalized Syt, respectively. Zooms represent typical boutons. Scale bars: 5 μm (full size) and 2 μm (zoom).
K   Quantifications of fluorescence intensity of internalized endogenous Syt at single presynaptic boutons of hippocampal neurons (DIV18) co-expressing RFP and pSuper empty vector, SCRN1 shRNA alone, or SCRN1 shRNA with GFP-SCRN1. *N* = 2, *n* = 201–300 boutons.

Data information: Data represent mean ± SEM; \*\*\**P* < 0.001, by Mann–Whitney *U*-test.
Source data are available online for this figure.

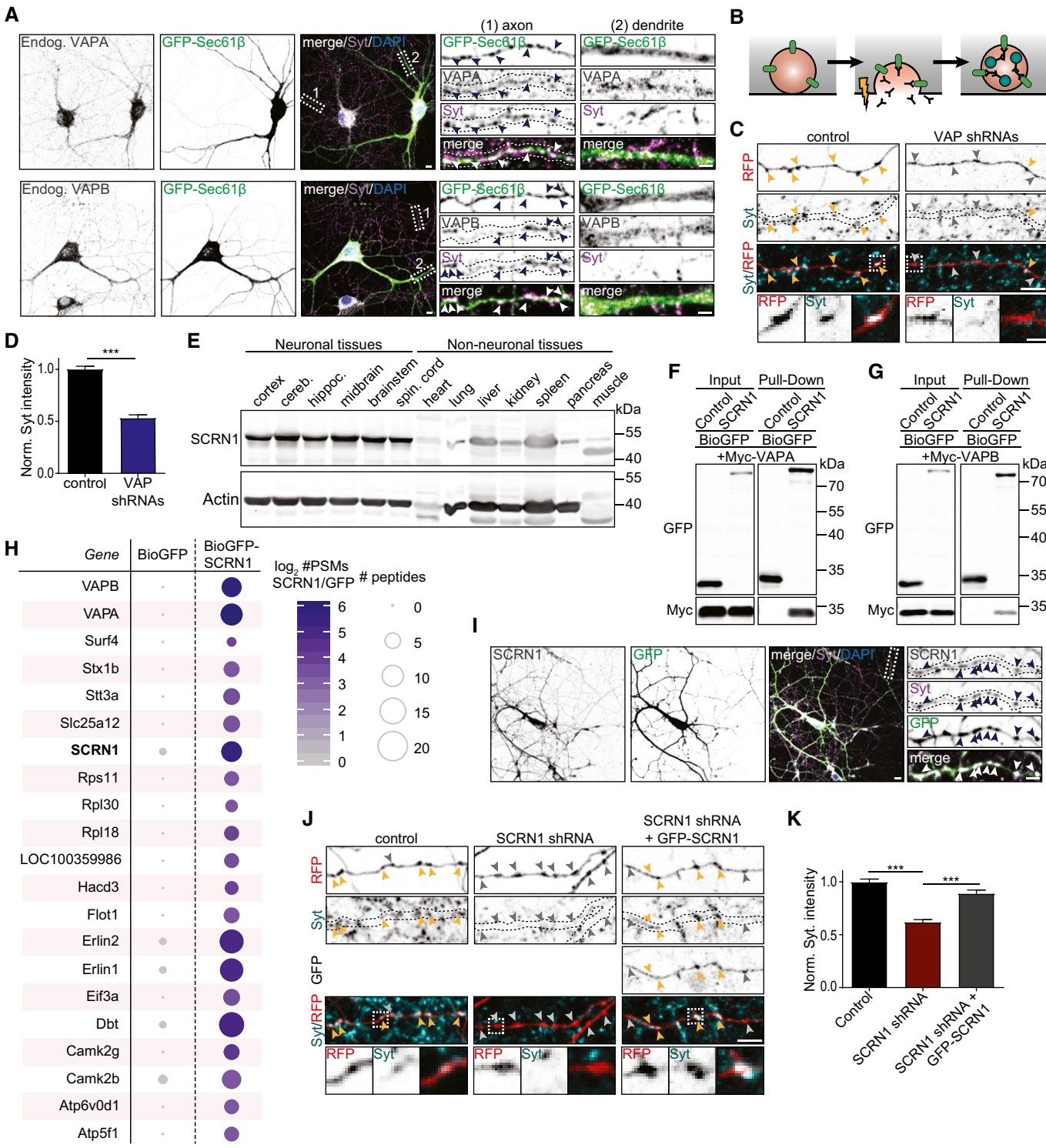

Figure 1.

whether VAPA and VAPB are engaged in modulating the SV cycle. This was addressed using the Syt antibody uptake assay, which provides a quantifiable read-out of exo- and endocytosis efficiency at presynaptic sites (Fig 1B). Live neurons were incubated with antibodies recognizing the luminal side of SV membrane protein Syt, while neurons were briefly stimulated by bicuculline.

Subsequently, neurons were fixed and the fluorescence intensity of internalized Syt at individual presynaptic boutons was measured. VAP was depleted from neurons by expressing shRNAs targeting VAPA and VAPB, which we have validated in the previous studies (Teuling *et al*, 2007; Kuijpers *et al*, 2013). Co-depletion of VAPA and VAPB showed a marked decrease (~ 50%) in Syt

internalization compared to control cells (Fig 1C and D). In addition, VAPA and VAPB knockdown resulted in a slight decrease in bouton size and bouton density (Figs 1C, and EV1B and C). In summary, we observed that loss of function of ER proteins VAPA and VAPB was accompanied by decreased SV cycling and defects in bouton maintenance.

## VAPA and VAPB associate with brain-enriched SCRN1 proteins

VAPs function as ER receptors for a large number of VAP-associated proteins containing a FFAT or FFAT-like motif (Murphy & Levine, 2016). To gain more insight into the underlying mechanism of VAP at presynaptic sites, we sought to identify the VAP interactor(s) that could be involved in controlling this function. In a recent study, many new VAP-associated proteins were identified by pull-down and mass spectrometry analysis, including the cytoplasmic protein SCRN1 (Murphy & Levine, 2016). Western blot analysis of lysates from different rat tissues using two different antibodies revealed that SCRN1 is abundantly enriched in brain tissues (Figs 1E and EV1D). This is consistent with the reported enriched expression of SCRN1 in the brain as described in various online expression databases (Protein Atlas, Expression Atlas, Alan Brain Atlas). We confirmed the association between VAP and SCRN1 with various biochemical assays. First, we conducted a pull-down experiment on lysates of HEK293T cells co-expressing biotinylated GFP (BioGFP) or GFP-SCRN1 (BioGFP-SCRN1) and Myc-VAPA or Myc-VAPB. Both Myc-VAPA and Myc-VAPB efficiently co-precipitated with BioGFP-SCRN1 (Fig 1F and G). Next, we examined the SCRN1 interactome using a more unbiased approach and performed BioGFP-SCRN1 pull-downs followed by mass spectrometry analysis using HEK293T cell lysates and adult rat brain extracts. The associations between SCRN1 and the VAPs were identified in both HEK293T lysates and brain extracts (Figs 1H and EV1E). Of all potential SCRN1-interacting proteins, both VAPA and VAPB showed the highest peptide-spectrum match (PSM) values in both datasets. Together, these biochemical data indicated that VAPs are associated with SCRN1.

## VAP-associated protein SCRN1 is involved in modulating SV cycling

We next tested whether SCRN1 was present at presynaptic sites and if this protein could be engaged in modulating SV cycling. Similar to VAP, immunostaining for endogenous SCRN1 revealed a punctate pattern throughout the neuron and regularly co-localized with presynaptic marker Syt (Fig 1I). Exogenous GFP-SCRN1 showed a diffuse cytoplasmic signal, which also co-localized with presynaptic boutons (Fig EV1F). To conduct loss-of-function experiments, we next generated and validated three shRNA targeting SCRN1 and continued our depletion experiments with a single shRNA (Fig EV1G–J). We tested the role of SCRN1 in SV cycling by conducting the Syt uptake assay in neurons depleted from SCRN1. SCRN1 knockdown also showed a marked decrease (~ 40%) in relative Syt internalization compared to control cells, thereby phenocopying the effect of VAP knockdown (Fig 1J and K). The presynaptic phenotype in SCRN1 knockdown neurons was rescued by expressing wild-type GFP-SCRN1 (Fig 1J and K). Together, these results illustrate that SCRN1 depletion, similarly to VAP depletion, results in impaired SV cycling.

## SCRN1 does not exhibit autolytic protease activity

To better understand the molecular function of VAP-associated protein SCRN1, we next tested whether its conserved proteolytic domain could be involved. Like all SCRN family members, SCRN1 contains a C69 protease domain and therefore belongs to the N-terminal nucleophile (Ntn) aminohydrolases superfamily (Pei & Grishin, 2003). Proteolytic activity in this superfamily relies on autolytic cleavage of the auto-inhibitory N-terminal of the precursor protein by the mature protein (Fig EV2A). This cleavage occurs right before the catalytic site of the protein, which is a cysteine residue in the SCRN family. Sequence alignment of the SCRN proteins revealed that the position of the predicted catalytic cysteine including the flanking residues is shared in SCRN2 and SCRN3, but not in SCRN1 (Fig EV2B). We analyzed N-terminal SCRN cleavage by conducting Western blotting of lysates from HEK293T cells expressing wild-type GFP-SCRN1, GFP-SCRN2, or GFP-SCRN3 (Fig EV2C). In lysates of GFP-SCRN2 and GFP-SCRN3 expressing cells, we identified a low molecular weight band corresponding to the predicted size of GFP fused to the N-terminal cleavage product. Conversely, this cleavage product was not observed in lysates of GFP-SCRN1 expressing cells. Moreover, mutant SCRN1, SCRN2, and SCRN3 constructs in which the predicted catalytic cysteine was replaced by a non-catalytic alanine residue also did not show a cleavage product (Fig EV2C). These data suggest that SCRN1, unlike its family members SCRN2 and SCRN3, does not exhibit autolytic protease activity.

## SCRN1 is recruited to VAP at the ER membrane

To further examine the function of VAP-associated protein SCRN1, we next assessed whether the subcellular localization of SCRN1 could be controlled by VAP. This was addressed by conducting co-expression experiments of GFP-SCRN1 and HA-VAPA or HA-VAPB in cultured neurons and COS7 cells. In COS7 cells, the ER structures are relatively less compact and easier to visualize than in neurons. GFP-SCRN1 expression alone in neurons or COS7 cells showed a diffuse cytoplasmic distribution, which only partly coincided with ER structures (Fig 2A and B). In neurons, co-expression of GFP-SCRN1 with either HA-VAPA or HA-VAPB resulted in the formation of dense VAP/SCRN1-positive clusters at neurites (Fig 2C). COS7 cells co-expressing GFP-SCRN1 and HA-VAPA or HA-VAPB showed marked differences in SCRN1 localization, as it induced abundant SCRN1 recruitment to VAP at the ER membrane (Fig 2D). This observation suggests that enhancing the number of VAPs at the ER membrane allows for increased SCRN1 binding, presumably because it decreases the competition with other FFAT(-like)-containing proteins for the VAP-binding pockets (Fig 2E). Next, we assessed whether the observed recruitment to VAP at ER structures is shared within the SCRN family. Contrarily, we observed no change in GFP-SCRN2 or GFP-SCRN3 localization when co-expressed with HA-VAPB in COS7 cells (Fig 2F). Together, these data indicate that SCRN1, and not SCRN2 and SCRN3, is recruited to VAP at the ER membrane.

## SCRN1 interacts with VAP through a single FFAT-like motif

Next, we sought to determine the specific domains responsible for the interaction between VAP and SCRN1. We found that the

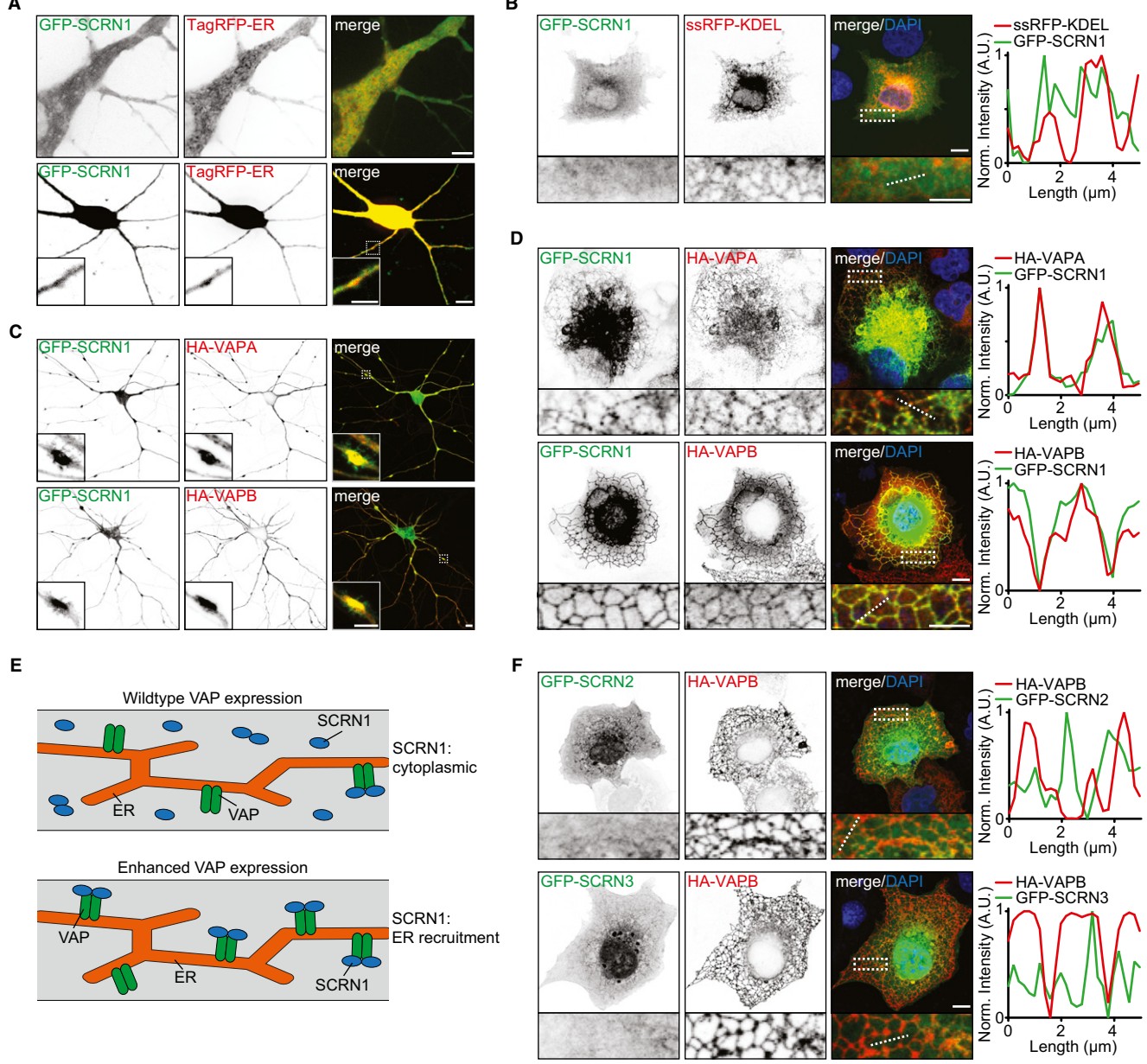

**Figure 2. SCRN1 is recruited to VAP at the ER membrane.**

A  Localization of GFP-SCRN1 and TagRFP-ER in hippocampal neurons (DIV16–18). Scale bars: 10 μm (bottom panel, full size) and 5 μm (top panel; bottom panel, zoom).
B  Localization of GFP-SCRN1 and ssRFP-KDEL in COS7 cells with normalized intensity plot of indicated line (dotted). Scale bars: 10 μm (full size) and 5 μm (zoom).
C  Hippocampal neurons (DIV16) co-expressing GFP-SCRN1 with HA-VAPA or HA-VAPB. Scale bars: 10 μm (full size) and 5 μm (zoom).
D  COS7 cells co-expressing GFP-SCRN1 with HA-VAPA or HA-VAPB with normalized intensity plot of indicated line (dotted). Scale bars: 10 μm (full size) and 5 μm (zoom).
E  Schematic illustration of SCRN1 recruitment to ER membranes upon increasing VAP levels.
F  COS7 cells co-expressing HA-VAPB with GFP-SCRN2 or GFP-SCRN3 with normalized intensity plot of indicated line (dotted). Scale bars: 10 μm (full size) and 5 μm (zoom).

C-terminal coiled-coil region of SCRN1 and the N-terminal major sperm protein (MSP) domain of VAPB are the minimal binding domains required for this interaction, as shown by co-expression experiments in COS7 cells and pull-down analysis of HEK293T lysates (Figs EV2D–I, 3A and E, and EV3B). The MSP domain of VAP contains a FFAT(-like) motif binding site, and FFAT(-like) motifs are found in the majority of the VAP-interacting proteins (Loewen *et al*, 2003; Murphy & Levine, 2016). Indeed, we found that the FFAT

binding motif in VAP is responsible for the interaction with SCRN1. The VAP mutant VAP-K87D/M89D, in which FFAT binding is disrupted, was no longer able to recruit GFP-SCRN1 (Fig 3A, B and E; Kaiser *et al*, 2005). Next, we searched for FFAT(-like) motifs in SCRN1 using a previously reported algorithm and identified four potential FFAT-like motifs (Fig EV3A; Murphy & Levine, 2016). We generated SCRN1 constructs with single-point mutations for each single FFAT-like motif (Fig 3A). VAP association was disrupted

when mutating the most C-terminal FFAT-like motif in SCRN1 (GFP-SCRN1-F402A), but not the other motifs, as shown by pull-down assays and co-expression experiments (Figs 3C–F, and EV3B–D).

Sequence alignment of the SCRN family members revealed that this newly identified FFAT-like motif in SCRN1 is not shared with the other two SCRN family members, which is consistent with our

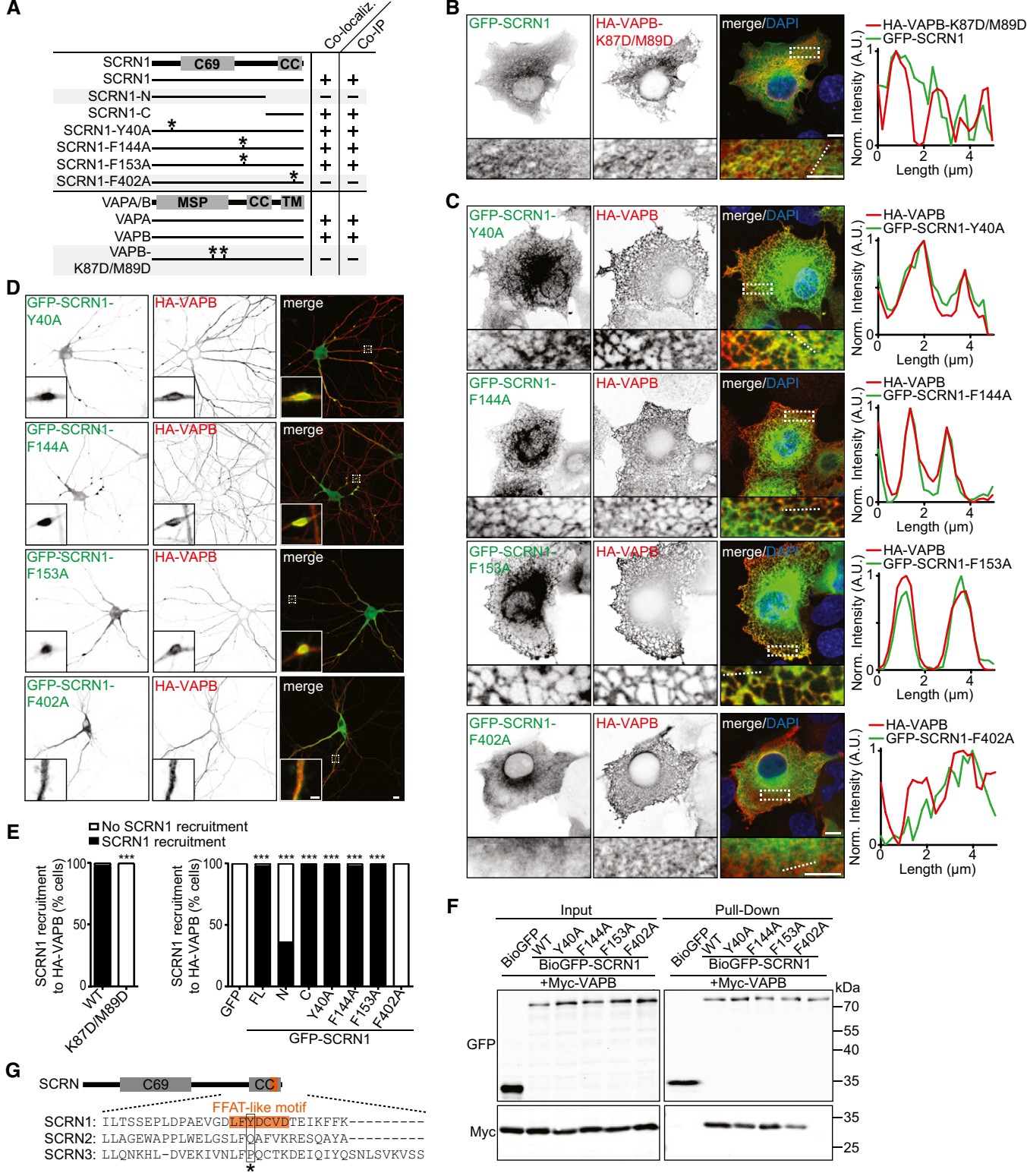

**Figure 3.**

**Figure 3.  VAP–SCRN1 interaction at the ER is mediated by a single FFAT-like motif.**

A   Schematic overview of SCRN1 and VAP constructs (asterisks represent mutations). Indicated is if expressed VAP/SCRN1 proteins co-localize in both hippocampal neurons and COS cells, and co-precipitate in pull-down assays.

B   COS7 cells co-expressing GFP-SCRN1 with HA-VAPB-K87D/M89D. Normalized intensity plot represents indicated line (dotted). Scale bars: 10 μm (full size) and 5 μm (zoom).

C   COS7 cells co-expressing HA-VAPB with GFP-SCRN1-Y40A, GFP-SCRN1-F144A, GFP-SCRN1-F153A, or GFP-SCRN1-F402A. Normalized intensity plot represents indicated line (dotted).Scale bars: 10 μm (full size) and 5 μm (zoom).

D   Hippocampal neurons (DIV16) co-expressing HA-VAPB with GFP-SCRN1-Y40A, GFP-SCRN1-F144A, GFP-SCRN1-F153A, or GFP-SCRN1-F402A. Scale bars: 10 μm (full size) and 5 μm (zoom).

E   Quantifications of SCRN1 recruitment to VAPB-positive structures in COS7 cells (%). Left graph: co-expression of GFP-SCRN1 with HA-VAPB or HA-VAPB-K87D/M89D (N = 2–3, n = 44–46). Right graph: co-expression of HA-VAPB with GFP or GFP-SCRN1-FL, GFP-SCRN1-N, GFP-SCRN1-C, GFP-SCRN1-Y40A, GFP-SCRN1-F144A, GFP-SCRN1-F153A, or GFP-SCRN1-F402A (N = 2–3, n = 41–64).

F   Pull-down assay of HEK293T cells co-expressing Myc-VAPB with GFP, GFP-SCRN1-WT, GFP-SCRN1-Y40A, GFP-SCRN1-F144A, GFP-SCRN1-F153A, or GFP-SCRN1-F402A.

G   Sequence alignment of the C-terminal human SCRN1, SCRN2, and SCRN3. Amino acid position 3 (asterisk) in FFAT-like motif (orange) of SCRN1 is not shared in SCRN2 and SCRN3.

Data information: ***$P < 0.001$, by chi-square test with *post hoc* analysis including Bonferroni correction.
Source data are available online for this figure.

observation that exogenous VAP is unable to recruit GFP-SCRN2 and GFP-SCRN3 (Figs 2F and 3G). Together, these data show that the MSP domains of VAPA and VAPB interact with a single FFAT-like motif in the C-terminal region of SCRN1.

### SCRN1 and VAP are required for proper ER morphology

Previously, it was shown that VAP interactions with FFAT-containing proteins are engaged in maintaining ER morphology (Kaiser *et al*, 2005). Next, we aimed to investigate whether its interaction with SCRN1 at the ER membrane could be important for this function. Similar to previously reported results, we found that COS7 cells expressing VAP-K87D/M89D, VAP mutant lacking proper FFAT binding, showed a robust phenotype that was characterized by non-reticular VAP structures throughout the cytosol (Figs 3B, 4A, and EV4A; Kaiser *et al*, 2005). Likewise, similar aberrant VAP-positive structures were found in COS7 cells expressing wild-type HA-VAP and GFP-SCRN1-F402A or GFP-SCRN1-N, both lacking a functional FFAT-like motif (Figs 3C, EV3C, 4A, and EV4A and B). These observed phenotypes were also detected with ER luminal marker TagRFP-ER, indicating that the non-reticular VAP signals represent affected underlying ER structures (Fig EV4C and D). Similarly, both in VAP and in SCRN1 knockdown neurons, as well as in neurons expressing GFP-SCRN1-F402A, ER morphology was severely disrupted (Fig 4B). Notably, in these neurons we observed dense ER patches surrounded by less dense or absent ER structures. On the other hand, neurons overexpressing wild-type SCRN1 also showed dense ER patches; however, these did seem properly connected to the rest of the ER structure. Thus, structural disruptions in ER morphology were consistently observed when the VAP–SCRN1 interactions were abrogated (Fig 4A). As such, expression of SCRN1-F402A showed the same phenotype on ER morphology as SCRN1 depletion, suggesting that SCRN1-F402A may act as a dominant-negative. As SCRN1-F402A is cytoplasmic, it could recruit and capture endogenous SCRN1 to the cytoplasmic pool, thereby preventing it from binding to VAP and execute its function at the ER. Indeed, oligomerization is a common feature of the Ntn amino-hydrolases superfamily, and pull-down assays showed that both SCRN1 and SCRN1-F402A were associated with other SCRN1 proteins (Fig EV4E and F). These results confirm that SCRN1 undergoes oligomerization and that SCRN1-F402A could act as a dominant-negative. Together, these data indicate that both SCRN1 and

VAP are required for proper ER morphology, which is mediated by VAP–SCRN1 interactions at the ER membrane.

### SCRN1 and VAP are engaged in maintaining ER continuity and dynamics

The ER structure undergoes constant remodeling while remaining continuous for proper functioning. Thus, next we sought to determine the effect of VAP and SCRN1 on both ER continuity and dynamics. We used live-cell imaging to map ER dynamics in cells expressing luminal ER marker TagRFP-ER. We observed fast remodeling of ER structures in both neurons and COS7 cells, ranging from ER tubule growth events and structural ER "wiggling" events (Figs 4C and D, and EV4G). The dense ER patches observed when expressing wild-type SCRN1 represented stabilized ER structures (Figs 4C and D, and EV4G; Movies EV1 and EV2). On the other hand, decreasing SCRN1 or VAP levels, or expressing dominant-negative SCRN1-F402A mutant, resulted in overall impaired dynamics of the dense ER patches that seemed partially discontinuous with the remaining ER structures (Figs 4C and D, and EV4G; Movie EV3). To gain more detailed insights into the role of VAP-SCRN1 interactions on ER morphology, we next sought to visualize ER nanostructures in neurons using the recently developed expansion microscopy (ExM) technique (Tillberg *et al*, 2016). This ExM approach allows for a ~ 4.5-fold physical sample magnification by isotropic chemical expansion and has been validated to preserve the nanoscale organization within different biological specimens (reviewed in Wassie *et al*, 2019). Here, we successfully resolved the dense neuronal ER structures, which enabled us to distinguish individual ER tubules and sheets in neurons expressing ER membrane marker GFP-Sec61β (Movie EV4; Figs 4E, and EV4H and I). Consistent with reported EM studies, we observed that the axonal ER network was comprised of 1 or 2 ER tubules per diameter and regularly formed tubular structures (Fig 4D; Terasaki, 2018; Wu *et al*, 2017; Yalcin *et al*, 2017). VAP or SCRN1 knockdown neurons showed marked differences on ER nanostructures. More specifically, these neurons showed severe discontinuity of ER tubules in axons, and overall less dense and irregular ER structures in axons, dendrites, and soma (Figs 4E, and EV4H and I). To quantitatively validate the role of VAP–SCRN1 interactions on ER continuity and dynamics, we conducted FRAP experiments on the previously observed characteristic dense ER patches in TagRFP-ER expressing

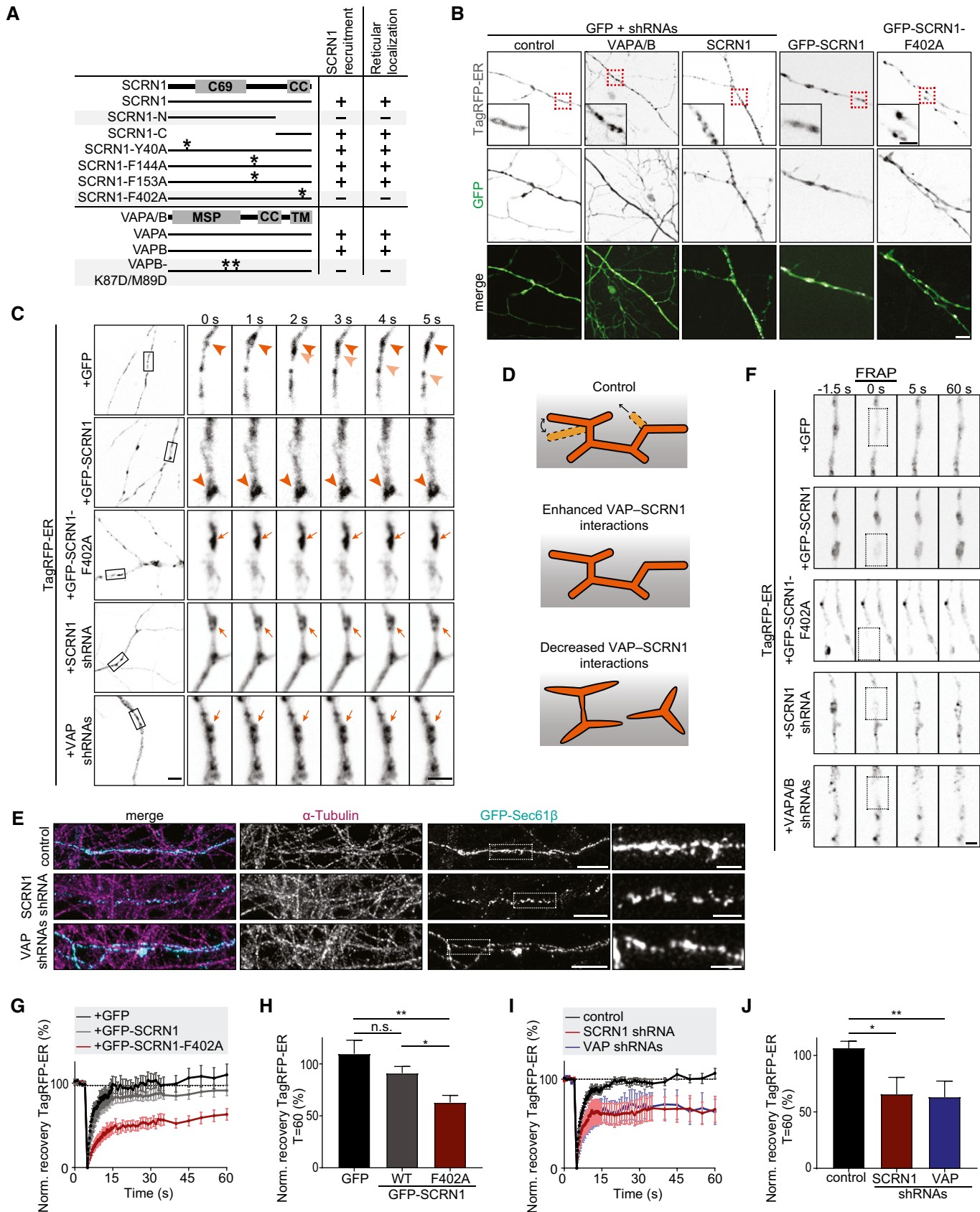

Figure 4.

**Figure 4. VAP and SCRN1 control ER continuity and remodeling.**

A   Summary of observed phenotypes on SCRN1 recruitment and ER morphology in cells co-expressing indicated SCRN1 and VAP constructs (asterisks represent mutations).

B   Live hippocampal neurons (DIV17–18) co-expressing TagRFP-ER with GFP, GFP-SCRN1, GFP-SCRN1-F402A, SCRN1 shRNA, or VAPA/B shRNAs. Scale bars: 5 μm (full size) and 2 μm (zoom).

C   Time-lapse of TagRFP-ER dynamics in hippocampal neurons (DIV16–18) co-expressing GFP, GFP-SCRN1, GFP-SCRN1-F402A, SCRN1 shRNA, or VAPA/B shRNAs. Intact and stable ER structures (dark arrowheads), intact and dynamic ER structures (light arrowheads), and impaired non-dynamic ER structures (arrows) are indicated. Scale bars: 5 μm (full size) and 2 μm (zoom).

D   Schematic illustration of the effect of VAP–SCRN1 interactions on ER structure and dynamics.

E   ER nanostructures visualized with GFP-Sec61β in axons of hippocampal neurons (DIV18) immunostained for α-tubulin and co-expressed with pSuper empty vector, SCRN1 shRNA, or VAPA/B shRNAs, and subjected to expansion microscopy. Scale bars: 2 μm (full size) and 500 nm (zoom).

F   FRAP experiment of TagRFP-ER in hippocampal neurons (DIV17–18) co-expressing GFP, GFP-SCRN1, and GFP-SCRN1-F402A. Scale bar: 2 μm.

G   Average normalized fluorescent TagRFP-ER recovery after photo-bleaching in hippocampal neurons (DIV17–18) co-expressing GFP, GFP-SCRN1, and GFP-SCRN1-F402A. $N = 2$, $n = 9$ neurons.

H   Normalized fluorescent TagRFP-ER recovery after photo-bleaching at $T = 60$ s in hippocampal neurons (DIV17–18) co-expressing GFP, GFP-SCRN1, and GFP-SCRN1-F402A. $N = 2$, $n = 9$ neurons.

I   Average normalized fluorescent TagRFP-ER recovery after photo-bleaching in hippocampal neurons (DIV18) co-expressing pSuper empty vector, SCRN1 shRNA, and VAPA/B shRNAs. $N = 4$, $n = 6$–$12$ neurons.

J   Normalized fluorescent TagRFP-ER recovery after photo-bleaching at $T = 60$ s in hippocampal neurons (DIV18) co-expressing pSuper empty vector, SCRN1 shRNA, and VAPA/B shRNAs. $N = 4$, $n = 6$–$12$ neurons.

neurons. Consistent with previous indications, photo-bleached TagRFP-ER co-expressed with either GFP or GFP-SCRN1 showed full recovery within 60 s of the ER patches (Fig 4F–H). This suggests rapid and complete redistribution of luminal ER content consistent with an intact ER structure. In contrast, the recovery of TagRFP-ER at ER patches was markedly reduced by ~ 45% in neurons expressing either GFP-SCRN-F402A, SCRN1 shRNA, or VAPA/B shRNAs, which implies incomplete redistribution of the luminal ER marker due to discontinuity (Fig 4F, I, and J). Together, these data indicate that loss of VAP and SCRN1 results in discontinuous ER structures and impaired ER dynamics in neurons, which is mediated by VAP-SCRN1 interactions.

## SCRN1-VAP interaction controls SV cycling

Next, we aimed to test whether the VAP–SCRN1 interactions at the ER membrane may be engaged in regulating the SV cycle, as we previously observed that VAP or SCRN1 depletion resulted in decreased SV cycling (Fig 1B–D, J and K). To test this, we conducted the Syt uptake assay in neurons depleted from SCRN1 and expressing SCRN1-F402A, thereby abolishing VAP–SCRN1 interactions. Unlike wild-type SCRN1 expression, exogenous mutant SCRN1-F402A was unable to rescue the effect of SCRN1 knockdown on SV cycling (Figs 5A and B). These data imply that the VAP–SCRN1 interactions, which we identified as regulators of ER remodeling, are engaged in modulating SV cycling at presynaptic sites.

## VAP and SCRN1 are engaged in regulating evoked presynaptic Ca²⁺ responses

Neurotransmitter release is induced by a local cytoplasmic $Ca^{2+}$ influx upon neuronal stimulation. Thus, to further investigate the phenotype of VAP and SCRN1 on SV cycling, we next assessed whether they are engaged in regulating evoked $Ca^{2+}$ influx. To test this, we measured $Ca^{2+}$ dynamics at single boutons with the genetically encoded $Ca^{2+}$ indicator GCaMP6f and triggered trains of action potentials (50 APs, 20 Hz) using electric field stimulation (Fig 5C). Presynaptic boutons were identified as GCaMP6f-positive swellings

along axons, which were shown to co-localize with the presynaptic active zone marker RIM1a-mCherry (Fig 5C). Loss of VAP or SCRN1 both resulted in a marked ~ 25% decreased peak amplitude of evoked $Ca^{2+}$ transients (Fig 5D–F). These results indicate that both the ER receptor VAP and the VAP-interacting protein SCRN1 are involved in modulating presynaptic $Ca^{2+}$ influx and thereby could affect SV cycling.

## VAP–SCRN1 interactions modulate Ca²⁺ homeostasis at presynaptic sites

To better understand the effects of VAP and SCRN1 on evoked $Ca^{2+}$ influx, we next sought to determine whether these proteins could interfere with ER-mediated $Ca^{2+}$ homeostasis. Maintaining basal $Ca^{2+}$ levels is one of the key functions of smooth ER, which is the only type of ER that is present in axons. Thus, we hypothesized that the observed structural ER defects with abolished VAP–SCRN1 interactions could perturb ER-mediated $Ca^{2+}$ homeostasis and thereby affect $Ca^{2+}$-mediated SV cycling. We compared relative basal $Ca^{2+}$ levels at presynaptic sites in neurons expressing $Ca^{2+}$ indicators GCaMP6f or R-GECO1, as well as mRFP or GFP to identify transfected cells and presynaptic boutons. Relative basal $Ca^{2+}$ levels were obtained by determining the ratio ($F_0/F_{max}$) of fluorescent GCaMP6f or R-GECO1 signals before ($F_0$) and after ($F_{max}$) ionomycin treatment. Ionomycin is an ionophore which induces $Ca^{2+}$ permeability at membranes, thereby enables effectively saturating $Ca^{2+}$ indicators to determine $F_{max}$. In VAP knockdown neurons, relative basal $Ca^{2+}$ levels at single boutons were markedly elevated (~ 2.5-fold) compared to control (Fig 5G–I). This is in line with the observed decreased evoked $Ca^{2+}$ influx, as higher basal $Ca^{2+}$ levels will result in a lower extracellular-cytoplasmic $Ca^{2+}$ concentration gradient. Consistently, presynaptic $Ca^{2+}$ levels were significantly increased (~ 2-fold) with dominant-negative SCRN1-F402A expression, but not with SCRN1 wild-type expression, indicating that the observed effect is mediated by VAP–SCRN1 interactions (Fig 5J–L). Together, these data imply that ER-localized VAP–SCRN1 interactions are engaged in modulating basal $Ca^{2+}$ levels at presynaptic sites.

# Discussion

The dynamic and continuous ER network extends throughout the axon, and evidence for its role in controlling presynaptic neurotransmitter release begins to emerge (Summerville *et al*, 2016; De Gregorio *et al*, 2017; de Juan-Sanz *et al*, 2017). In this study, we identified novel control mechanisms for ER remodeling, presynaptic $Ca^{2+}$ homeostasis, and $Ca^{2+}$-induced SV cycling, which are mediated by ER receptor VAP together with VAP-interacting protein SCRN1. Together, these data point toward a model where VAP–SCRN1 interactions tune ER integrity and dynamics, and thereby could modulate basal $Ca^{2+}$ levels and subsequently SV cycling at presynaptic sites.

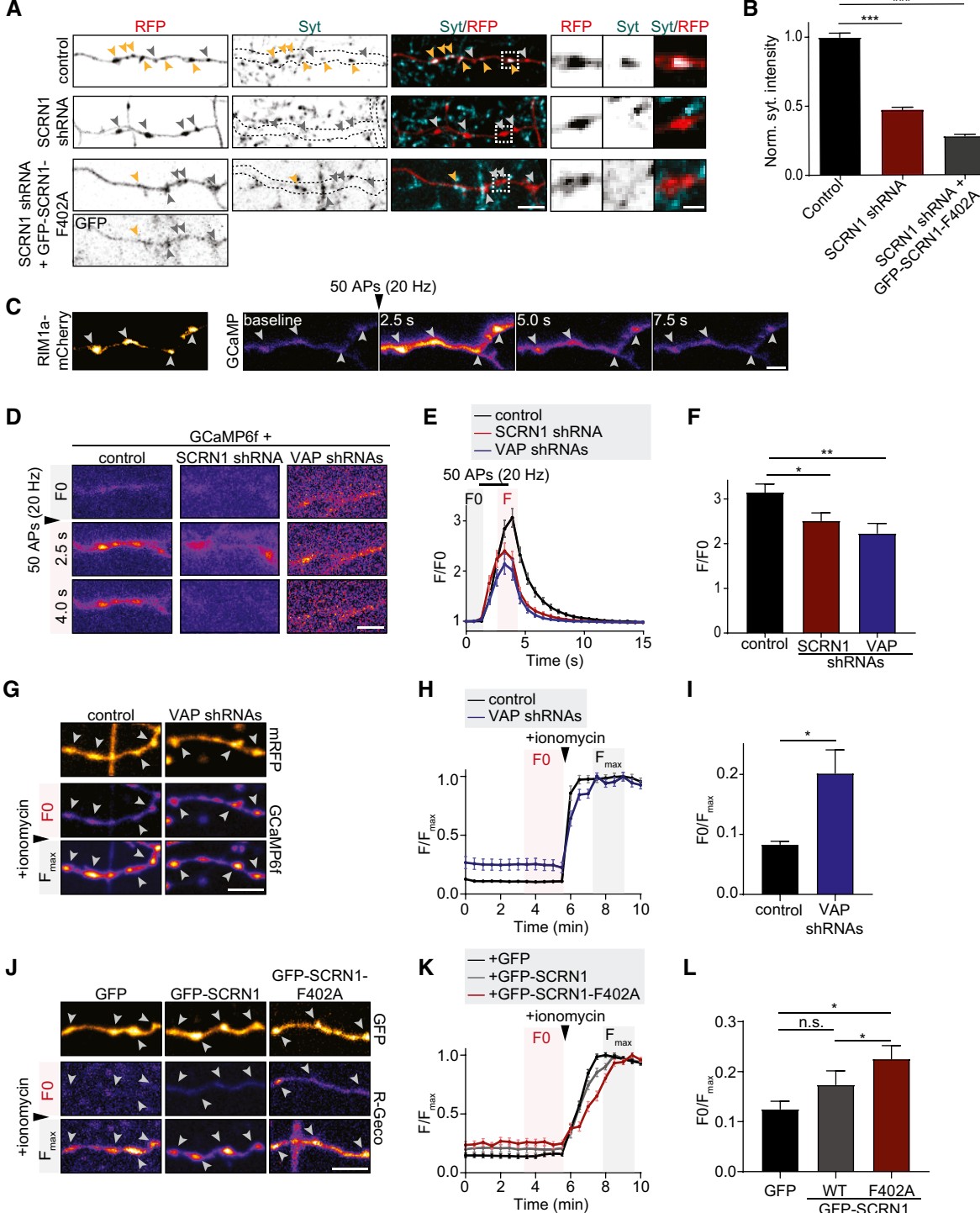

**Figure 5.**

**Figure 5.  VAP and SCRN1 control Ca²⁺ homeostasis and presynaptic release.**

A   Representative image of Syt antibody uptake at axons of hippocampal neurons (DIV18) co-expressing RFP and pSuper empty vector, SCRN1 shRNA, or SCRN1 shRNA with GFP-SCRN1-F402A. Yellow and gray arrowheads mark presynaptic boutons with and without internalized Syt, respectively. Zooms represent typical boutons. Scale bars: 5 μm (full size) and 2 μm (zoom).

B   Quantifications of fluorescence intensity of internalized endogenous Syt at individual presynaptic boutons of hippocampal neurons (DIV18) co-expressing RFP and pSuper empty vector, SCRN1 shRNA, or SCRN1 shRNA with GFP-SCRN1-F402A. $N = 2$, $n = 201$–300 boutons.

C   Representative time-lapse of cytosolic GCaMP6f upon electric field stimulation (50 APs, 20 Hz) in axons of hippocampal neurons (DIV21) co-expressed with RIM1a-mCherry to visualize presynaptic sites (arrowheads). Scale bar: 5 μm.

D   Time-lapses of cytosolic GCaMP6f upon electric field stimulation (50 APs, 20 Hz) in axons of hippocampal neurons (DIV21) co-expressing pSuper control, SCRN1 shRNA, or VAPA/B shRNAs. Scale bar: 5 μm.

E   Average normalized response of GCaMP6f fluorescent intensity at presynaptic boutons upon stimulation (50 APs, 20 Hz) in hippocampal neurons (DIV21) co-expressing pSuper empty vector, SCRN1 shRNA, or VAPA/B shRNAs. $N = 5$–6, $n = 15$–27.

F   Average normalized peak response of GCaMP6f at presynaptic boutons upon stimulation (50 APs, 20 Hz) in hippocampal neurons (DIV21) co-expressing pSuper empty vector, SCRN1 shRNA, or VAPA/B shRNAs. $N = 5$–6, $n = 15$–27.

G   Representative time-lapse of cytosolic GCaMP6f before ($F_0$) and after ($F_{max}$) ionomycin treatment at axons of hippocampal neurons (DIV18) co-expressed with mRFP and pSuper empty vector or VAPA/B shRNAs. Arrowheads mark presynaptic boutons. Scale bar: 5 μm.

H   Basal GCaMP6f fluorescence ($F$) normalized to the maximum GCaMP6f fluorescence ($F_{max}$) after ionomycin treatment at presynaptic boutons of hippocampal neurons (DIV18) co-expressing mRFP with pSuper empty vector or VAPA/B shRNAs. $N = 2$, $n = 47$–50.

I   Average basal GCaMP6f fluorescence ($F_0$) normalized to the max GCaMP6f fluorescence intensity ($F_{max}$) after ionomycin treatment at presynaptic boutons of hippocampal neurons (DIV18) co-expressing mRFP with pSuper empty vector or VAPA/B shRNAs. $N = 2$, $n = 47$–50.

J   Representative time-lapse of cytosolic R-GECO1 before ($F_0$) and after ($F_{max}$) ionomycin treatment at axons of hippocampal neurons (DIV18) co-expressing GFP, GFP-SCRN1, or GFP-SCRN1-F402A. Arrowheads mark presynaptic boutons. Scale bar: 5 μm.

K   Basal R-GECO1 fluorescence ($F$) normalized to the maximum R-GECO1 fluorescence ($F_{max}$) after ionomycin treatment at presynaptic boutons of hippocampal neurons (DIV18) co-expressing GFP, GFP-SCRN1, or GFP-SCRN1-F402A. $N = 3$, $n = 70$–89.

L   Average basal R-GECO1 fluorescence ($F_0$) normalized to the maximum R-GECO1 fluorescence intensity ($F_{max}$) after ionomycin treatment at presynaptic boutons of hippocampal neurons (DIV18) co-expressing GFP, GFP-SCRN1, or GFP-SCRN1-F402A. $N = 3$, $n = 70$–89.

Data information: Data represent mean ± SEM; n.s.: not significant; *$P < 0.05$; **$P < 0.01$; ***$P < 0.001$, by Mann–Whitney *U*-test.

## SCRN1 is a VAP-interacting protein

Here, we demonstrated that loss of ER membrane receptor VAP results in discontinuous ER structures, impaired ER dynamics, and decreased SV cycling. This is in line with previous studies which already hinted for a role of VAP in maintaining ER morphology and regulating synaptic function (Skehel *et al*, 1995; Kaiser *et al*, 2005; Gomez-Suaga *et al*, 2019). To gain further mechanistic insights into the VAP-mediated phenotypes, we sought to identify a VAP-associated protein involved in this function. Hence, we selected SCRN1 as a potential candidate for several reasons. First, SCRN1 was abundantly expressed in brain tissue, as indicated by various expression databases (Protein Atlas, Expression Atlas, Alan Brain Atlas) and confirmed by our Western blot analysis of different rat tissues. Second, in non-neuronal cells, SCRN1 was found to play a role in regulating Ca²⁺-controlled exocytosis, which is also a key process of the SV cycle (Way *et al*, 2002; Lin *et al*, 2015). We confirmed the interaction between VAP and SCRN1, and identified a single FFAT-like motif responsible for the interaction. Consistent with our hypothesis, we found that either SCRN1 depletion or dominant-negative SCRN1-F402A expression phenocopied the effect of VAP depletion on both ER remodeling and SV cycling. The dominant-negative effect was consistently more potent than the shRNA knockdown effect, similar to what is often observed when comparing dominant-negative and shRNA silencing expression constructs of proteins. Here, the difference could be explained by incomplete depletion of endogenous SCRN1 during the SCRN1 shRNA silencing period, whereas on the other hand dominant-negative SCRN1-F402A actively recruits and captures endogenous SCRN1 proteins and thereby impairs its function. Nevertheless, we cannot exclude that additional functions of SCRN1-F402A might be at play. Together, our data indicate that SCRN1-VAP interactions are engaged in controlling the

VAP-associated phenotypes, and further investigations are required to examine whether additional VAP-interacting proteins could also be involved in modulating SV cycling.

## VAP–SCRN1 interactions control ER continuity and dynamics

The ER network is composed of a well-maintained interconnected structure that undergoes continuous remodeling for proper functioning. Here, we identified the VAP–SCRN1 interactions as critical regulators for maintaining ER structure and dynamics. Considering that VAPs are best known to act as major ER receptors by facilitating tight membrane contact sites between the ER and other organelles, it is tempting to speculate that these membrane contact sites might be engaged in controlling VAP-mediated functions on ER integrity and dynamics (Muallem *et al*, 2017; Wu *et al*, 2018). One plausible mechanism could be that these membrane contact sites may act as "anchor points" which locally stabilize the ER network. These VAP-facilitated membrane contact sites are abundantly present throughout the cell and can thus stabilize local spots within the large interconnected and highly dynamic ER network. As such, this hypothesis follows the assumption that proper ER integrity relies on a defined balance between ER dynamics and stability. In line with this, it was previously shown that depleting either ER-forming protein atlastin or ER-stabilizing protein reticulon resulted in ER tubule fragmentation, which could remarkably be rescued by depleting both proteins simultaneously (Wang *et al*, 2016). Alternatively, VAP-mediated membrane contact sites could control ER remodeling by enabling fast lipid delivery. We showed that ER remodeling in neurons is a fast process and that ER tubule growth events can occur in a few seconds. This fast ER remodeling requires the continuous and rapid rearrangement of lipids. This may be accomplished via fast lipid transfer at membrane contact sites between ER and

other organelles, rather than via the relatively slower lipid synthesis and redistribution at the ER membrane. Note worthily, considering that VAPs are implicated in a wide range of functions, various other possible direct or indirect VAP-mediated mechanisms to control ER remodeling might be at play.

## Maintaining axonal ER structures is important to preserve presynaptic function

In recent EM studies, it was reported that ER structures in axons show specific adaptations, characterized by small ER lumen and low density of 1–2 tubules per axon diameter. These adaptations likely allow the axonal ER to extend throughout all axon branches while remaining continuous with the remaining ER network, which is typically more dense and branched. In this study, we found that VAP and SCRN1 depletion affected ER structures throughout the cell, though axonal ER structures seemed most severely disrupted. Interestingly, mutations in generic structural ER proteins such as atlastin-1, reticulon-2, REEP1, and REEP2 are causative for the axonopathy HSP (Hazan *et al*, 1999; Zhao *et al*, 2001; Nishimura *et al*, 2004; Zuchner *et al*, 2006; Montenegro *et al*, 2012; Esteves *et al*, 2014; Yalcin *et al*, 2017). Together, these findings indicate that the delicate ER network in axons is more sensitive to maintenance defects. Therefore, ubiquitous ER disruptions may result in more profound phenotypes on particularly axonal functions. Here, we demonstrated that the observed ER abrogations with decreased VAP–SCRN1 interactions were accompanied with impaired SV cycling at presynaptic sites. Consistent with this, recent studies in *Drosophila* neurons showed that loss of ER-shaping protein atlastin or reticulon also resulted in impaired ER structures as well as decreased neurotransmitter release (Summerville *et al*, 2016; De Gregorio *et al*, 2017). Together, these findings imply that ER structure and function are engaged in modulating SV cycling. Notably, investigating the direct relation between these phenotypes is challenging, as it is not feasible to specifically isolate the local function of ER at presynaptic sites since all ER membranes and lumen in the cell are continuous. Nevertheless, considering that the neuronal ER network remains fully continuous despite its complex cellular morphology may actually highlight the functional relevance of this continuity for its role at presynaptic sites. Toward this end, it would be interesting to direct future work on investigating whether the continuity of the neuronal ER network is important for the reported ER-mediated functions at presynaptic sites.

## VAP–SCRN1 interactions modulate presynaptic $Ca^{2+}$ dynamics and SV cycling

In this study, we reported that the ER defects observed with VAP or SCRN1 depletion are accompanied with reduced presynaptic $Ca^{2+}$ influx and SV cycling. Considering that axonal ER structures are exclusively comprised of smooth ER, we hypothesized that defects in maintaining $Ca^{2+}$ homeostasis, which is a key function of smooth ER, could provide a mechanistic link between the observed phenotypes on ER integrity and SV cycling. Previous reports already hinted for a correlation between ER-mediated $Ca^{2+}$ homeostasis and neurotransmitter release. In a recent study, a feedback loop between ER $Ca^{2+}$ concentration, presynaptic $Ca^{2+}$ influx, and SV exocytosis was identified in dissociated rodent neurons (de Juan-Sanz *et al*,

2017). Additionally, in *Drosophila* HSP models, both impaired ER integrity and affected SV cycling were rescued upon $Ca^{2+}$ bath application (Summerville *et al*, 2016). In this report, we found that loss of VAP–SCRN1 interactions in neurons results in elevated basal $Ca^{2+}$ levels at presynaptic sites. This implies that the cytoplasmic-extracellular $Ca^{2+}$ concentration gradient is reduced, which could explain the reduction in evoked $Ca^{2+}$ response and subsequent decreased SV cycling. Consistently, previous reports in non-neuronal cells showed that VAP-mediated membrane contact sites regulate $Ca^{2+}$ homeostasis and that SCRN1 controls $Ca^{2+}$-dependent processes (Way *et al*, 2002; Lin *et al*, 2015; Paillusson *et al*, 2017). It remains poorly understood how prolonged increase in basal $Ca^{2+}$ levels leads to reduced SV cycling. We speculate that chronic elevation of basal $Ca^{2+}$ levels could result in compensatory responses that may lead to downscaled synaptic strength. This could be accomplished at different levels, e.g., by decreasing bouton size, lower number of synapses and SVs, and downregulation of proteins involved in the SV cycle machinery. Consistent with this idea, we observed less and smaller boutons in VAP knockdown neurons. In addition to the $Ca^{2+}$-mediated effects, the smooth ER in axons could also modulate SV cycling by controlling lipid homeostasis. The presynaptic membrane is comprised of a unique presynaptic lipid composition that is required for proper overall presynaptic function (Lauwers *et al*, 2016). Possibly, this presynaptic lipid composition could be facilitated and maintained by enabling lipid delivery at VAP-mediated membrane contact sites. Taken together, it would be interesting to direct future research in exploring the possible roles of VAP–SCRN1 interactions in controlling $Ca^{2+}$ homeostasis and presynaptic lipid composition, and on how this could modulate the tightly spatiotemporal controlled SV cycle.

## Molecular function of VAP-interacting protein SCRN1

It remains unclear how the SCRN1 interaction with VAP at the ER membrane may control the observed phenotypes on ER integrity. The C69 protease domain of SCRN1 did not show proteolytic activity, whereas this was observed for family members SCRN2 and SCRN3. Thus, it is unlikely that the observed VAP-mediated functions involve enzymatic activity of SCRN1. However, we did observe oligomerization of SCRN1, which may hint for a scaffolding function of SCRN1. Possibly, as scaffolding protein, SCRN1 could promote stabilization of VAP interactions at membrane contact sites. Consistent with this, we observed increased stabilization of ER structures upon wild-type SCRN1 expression. This highlights the importance of controlling the endogenous levels of SCRN1 at the ER membrane in order to balance between ER dynamics and stability. This is supported by the observation that endogenous SCRN1 proteins were not robustly localized at ER structures, whereas SCRN1 proteins were fully recruited to VAP at the ER membrane upon elevating VAP and SCRN1 levels. Similarly, it was previously shown that many other proteins containing a FFAT(-like) motif also did not fully coincide with ER structures (Murphy & Levine, 2016). Together, these findings imply that SCRN1, as well as many other FFAT(-like) proteins, undergoes continuous cycles of competitive binding and unbinding to the limited available VAP-binding pockets. As such, we propose that controlling endogenous intracellular SCRN1 levels, and thus competition with other VAP interactors, could be a novel mechanism to tune ER dynamics and subsequently

presynaptic function. Moreover, unlike the ubiquitously expressed VAPs, SCRN1 expression is highly enriched in brain tissue. Therefore, SCRN1 may be engaged in controlling VAP-mediated functions in brain tissue specifically.

In summary, we propose that VAP–SCRN1 interactions act as a novel control mechanism for dynamic ER remodeling, and consequently $Ca^{2+}$ homeostasis and SV cycling. Future work is required to better understand the molecular function of VAP–SCRN1 interactions in mediating ER integrity and $Ca^{2+}$-driven SV cycling. Finally, investigating additional ER-mediated control mechanisms that are engaged in modulating presynaptic function is required to obtain more insights into the precise function of the dynamic and continuous neuronal ER network in controlling SV cycling.

# Materials and Methods

## Animals

All animal experiments were approved by the Animal Ethical Review Committee (DEC) of Utrecht University and performed in accordance with the guidelines for the welfare of experimental animals issued by the Dutch law and following European regulations (Guideline 86/609/EEC).

## Primary rat neuron culture and transfection

Dissociated hippocampal and cortical neuron cultures were prepared from embryonic day 18 rat pups of mixed gender. Cells were plated on 18-mm glass coverslips coated with poly-L-lysine (37.5 mg/ml) and laminin (1.25 mg/ml) in a 12-well plate at a density of 100k/well for hippocampal neurons and 50k/well for cortical neurons. Cultures were maintained in Neurobasal medium (NB) supplemented with 2% B27, 0.5 mM glutamine, 16.6 μM glutamate, and 1% penicillin/streptomycin at 37°C in 5% $CO_2$.

Neuron cultures were transfected using a mixture of 3.3 μl lipofectamine 2000 (Invitrogen), 1.5–3.0 μg DNA, and 200 μl NB per coverslip. For knockdown experiments, a total of 1.5 μg DNA of shRNA construct(s) was used per coverslip. The transfection mixture was added to the coverslips, which were placed in fresh NB supplemented with 0.5 mM glutamine, and incubated for 45–90 min. Next, coverslips were washed once in prewarmed NB and transferred back to their original medium. Cells were maintained for 4 days (for knockdown experiments) or 24–48 h (for other experiments) prior to fixation or live-cell imaging.

## Cell line culture and transfection

COS7 and HEK293T cells were cultured on plastic at 37°C in 5% $CO_2$ in DMEM/Ham's F10 (50%/50%) medium supplemented with 10% FCS and 1% penicillin/streptomycin. COS7 and HEK293T cells were plated on, respectively, 18-mm glass coverslips or plastic 1 day prior to transfection with MaxPEI (Polysciences). In brief, MaxPEI/DNA (3:1 ratio) was mixed in fresh serum-free DMEM or Ham's F10 medium, incubated for 20 min, and added to the cell culture. After 24 h, cells were either processed for biochemistry, fixed, or used for live-cell imaging.

## DNA plasmids

The following plasmids are described previously: Myc-VAPA, Myc-VAPB, HA-VAPA (with E178G mutation), HA-VAPB, HA-VAPB-K87D/M89D, GFP-VAPB-TM, GFP-VAPB-MSP-CC, and VAPB-MSP-GFP (with E6I single nucleotide polymorphism and K139R mutation; Teuling *et al*, 2007); pGFPC1-Sec61β (Hradsky *et al*, 2011); HA-BirA (de Boer *et al*, 2003); BioGFP (Jaworski *et al*, 2009); GW1-RFP and pGW1-GFP (Hoogenraad *et al*, 2005); pSuper vector (Brummelkamp *et al*, 2002); ssRFP-KDEL (Addgene plasmid #62236, gift from Dr. Erik Snapp; Snapp *et al*, 2006), TagRFP-ER (Schatzle *et al*, 2018), HA-Erlin1, and HA-Erlin2 (gift from Dr. Richard J.H. Wojcikiewicz; Pearce *et al*, 2007, 2009); GCaMP6f (Addgene plasmid #58514, gift from Prof. Adam E. Cohen; Venkatachalam & Cohen, 2014); and R-GECO1 (Addgene plasmid #45494, gift from Prof. Robert E. Campbell; Wu *et al*, 2013).

The cDNAs of SCRN1a (AAH_40492.1), SCRN2 (AAH_10408.2), and SCRN3 (AAI_19685.1) were obtained from a human cDNA library kindly provided by Dr. Mike Boxem. All wild-type SCRN1, SCRN1-N (1–293), SCRN1-C (293–414), SCRN2, and SCRN3 constructs were generated using PCR-based cloning strategies and inserted into β-actin (for HA-SCRN1) or GW1 (for all other constructs) vectors. Constructs with single-point mutations were generated using site-directed mutagenesis. SCRN1 FFAT-like mutant constructs were obtained for each predicted FFAT-like motif identified by a previously reported algorithm (Murphy & Levine, 2016). More specifically, SCRN1-Y40A, SCRN1-F144A, SCRN1-F153A, and SCRN1-F402A were generated by replacing the conserved hydrophobic phenylalanine or tyrosine residue for an alanine residue. Proteolytic dead mutant constructs SCRN1-C9A, SCRN2-C12A, and SCRN3-C6A were generated by replacing the predicted proteolytic cysteine residue, as identified by the online MEROPS database, by a non-catalytic alanine residue. The RIM1a-mCherry construct was obtained by exchanging the HA-tag of the previously reported pAJ14063-pFUGW-RIM1aWT-HA construct (de Jong *et al*, 2018). The following shRNAs inserted in pSuper vectors were used in this study: VAPA shRNA #1 (5′-GCATGCAGAGTGCTGTTTC-3′; Teuling *et al*, 2007), VAPA shRNA #2 (5′-GGAAACTGATGGAAGAGTG-3′; Teuling *et al*, 2007), and VAPB shRNA #1 (5′-GGTGATGGAA GAGTGC-3′; Teuling *et al*, 2007); and SCRN1 shRNA #1 (5′-GATCCTTCCAGGTCCATAT-3′), SCRN1 shRNA #2 (5′-GCACTTA CATCTCAATTGA-3′), and SCRN1 shRNA #3 (5′-CAGGC TTGGTTTAGAACGA-3′).

## Antibodies

The following primary antibodies were used for this study: rabbit anti-VAPA (homemade #1006-04; Teuling *et al*, 2007) and anti-VAPB (homemade #1006-00; Teuling *et al*, 2007); mouse anti-synaptotagmin (SySy, 105311, clone 604.2); rabbit anti-SCRN1 (SySy, 289003; used in Fig 1E), rabbit anti-SCRN1 (Abcam, ab105355; used in Fig EV1D), and rabbit anti-SCRN1 (Sigma, HPA024517, RRID:AB_2184811; used for all other experiments; validation SCRN1 antibodies in Fig EV1G–J); guinea pig anti-vGlut (Millipore, ab5095); rat anti-HA (Roche, 1867423; used for immunostainings); mouse anti-HA (BioLegend, mms-101p, clone 16B12; used for immunoblots); mouse anti-actin (Chemicon, MAB1501R, clone C4); rabbit anti-GFP (Abcam, ab290); mouse

anti-Myc (Santa Cruz, SC40, clone 9E10); and mouse α-Tubulin (Sigma, T5168, clone B-5-1-2, RRID:AB_477579). The following secondary antibodies were used for this study: anti-rabbit Alexa 488 (Life Technologies, A11034), anti-rabbit Alexa 568 (Life Technologies, A11036), anti-rat Alexa 568 (Life Technologies, A11077), anti-guinea pig Alexa 568 (Life Technologies, A11075), anti-mouse Alexa 647 (Life Technologies, A21236), anti-mouse anti-HRP (Dako, P0260), anti-rabbit anti-HRP (Dako, P0399), anti-mouse IRDye 680LT (Li-Cor, 926-68020), and anti-rabbit IRDye 800CW (Li-Cor, 926-32211).

### Tissue extracts, cell extracts, and immunoblotting

To generate tissue extracts for Western blot and mass spectrometry analysis, different brain regions (cerebellum, cortex, hippocampus, midbrain, brainstem, and spinal cord) or whole brains were dissected from adult female rats. Samples were homogenized in ice-cold homogenization buffer (150 mM NaCl, 50 mM Tris, 0.1% SDS, 0.2% NP-40, pH 7.8) supplemented with 1× complete protease inhibitor cocktail (Roche), sonicated, and centrifuged (15 min, 900 $g$, 4°C). Protein concentrations of supernatant were measured using a BCA protein assay (Pierce). Next, 20 μg protein per sample was resuspended in SDS sample buffer and boiled for 5 min at 95°C. To generate cell extracts for Western blot analysis, transfected HEK293T cells were washed and harvested in ice-cold PBS. Cells were centrifuged (5 min, 300 $g$, 4°C), and the pellet was resuspended in ice-cold lysis buffer (100 mM Tris, 150 mM NaCl, 1% Triton, pH 7.5) supplemented with 1× complete protease inhibitor cocktail (Roche). Cell lysates were centrifuged (5 min, 20,000 $g$, 4°C), and supernatant was resuspended in SDS sample buffer and boiled for 10 min at 100°C. Samples were resolved on SDS–page gels and transferred to nitrocellulose membranes (Bio-Rad) or polyvinylidene difluoride membranes (Millipore). Membranes were blocked for 30 min with PBS-T (PBS with 0.05% Tween) with 2% BSA. Next, membranes were sequentially incubated with primary and secondary antibodies diluted in PBS-T with 2% BSA, and washed three times with PBS-T after each antibody incubation step. Proteins resolved on the membranes were visualized using Odyssey Infrared Imaging (Li-Cor Biosciences) or enhanced chemiluminescence.

### Pull-down assays and mass spectrometry analysis

For biotin–streptavidin pull-down assays, HEK293T cells were co-transfected with BirA, BioGFP(-fusion) plasmids (used as bait), and an additional plasmid (used as prey). After ~ 24 h, cells were washed once with ice-cold PBS, harvested in ice-cold PBS supplemented with 0.5× complete protease inhibitor cocktail (Roche), and centrifuged (5 min, 300 $g$, 4°C). Cell pellets were resuspended in ice-cold lysis buffer (100 mM Tris, 150 mM NaCl, 1% Triton, pH 7.5) supplemented with 1× complete protease inhibitor cocktail, incubated for 10 min on ice, and centrifuged (5 min, 20,000 $g$, 4°C). Supernatant was used for the binding assay and for generating input samples by boiling for 5 min at 100°C in SDS sample buffer. Beads were pretreated before the binding assay. For regular pull-down assays with cell culture extracts, magnetic Dynabeads M-280 Streptavidin (Thermo Fisher Scientific) were prewashed once with normal washing buffer (20 mM Tris HCl, 150 mM KCl, 0.5% Triton, pH 7.5), incubated for 30 min at room temperature with blocking

buffer (20 mM Tris, 150 mM KCl, 0.2 μg/μl CEA, pH 7.5), and washed twice with normal washing buffer. Binding of HEK293T cell lysates and beads was performed for 1 h at 4°C. Beads were subsequently washed five times using normal washing buffer and boiled for 5 min at 100°C in lysis buffer with SDS sample buffer to elute proteins and generate pull-down samples. Alternatively, for pull-down assays with whole brain extracts for mass spectrometry analysis, beads were prewashed twice with low salt buffer (100 mM KCl, 0.1% Triton X-100, 20 mM Tris, pH 7.6), twice with high salt buffer (500 mM KCl, 0.1% Triton X-100, 20 mM Tris, pH 7.6), and twice again in low salt buffer. Binding of HEK293T cell lysates and beads was performed for 1 h at 4°C in presence of whole rat brain extract (prepared as described above), and beads were subsequently washed five times using normal washing buffer. Mass spectrometry analysis of samples was conducted as described before (Cunha-Ferreira *et al*, 2018). All the mass spectrometry proteomics data have been deposited to the Pride database (http://www.ebi.ac.uk/pride) with the dataset identifier PXD014534.

### Immunofluorescence staining

Cells were fixed for 10 min in 4% formaldehyde and 4% sucrose (neurons) or in 4% formaldehyde (COS7 cells) at room temperature and washed three times with PBS. Fixed neurons were sequentially incubated with primary and secondary antibodies diluted in GDB (0.2% BSA, 0.8 M NaCl, 0.5% Triton X-100, 30 mM phosphate buffer, pH 7.4). Fixed COS7 cells were first permeabilized for 10 min in PBS with 0.1% Triton-X, blocked for 30 min in PBS with 2% BSA, and sequentially incubated with primary and secondary antibodies diluted in PBS with 2% BSA. Cells were washed three times with PBS after each antibody incubation step.

### Expansion microscopy sample preparation

Expansion microscopy was performed according to proExM protocol (Tillberg *et al*, 2016). Briefly, immunostained cells on 18-mm glass coverslips were incubated overnight in PBS with 0.1 mg/ml Acryloyl-X (Thermo Fisher, A20770) and 0.002% of 0.1 μm yellow–green Fluorospheres (Thermo Fisher, F8803). These bright fluorescent microspheres adhered to cell surfaces, thereby this cell boundary marker simplified the localization of cells in the expanded samples. Cells were washed three times with PBS and transferred to a gelation chamber (13 mm diameter, 120 μl volume) made of silicone molds (Sigma-Aldrich, GBL664107) on a parafilm-covered glass slide. The chamber was prefilled with monomer solution (PBS, 2 M NaCl, 8.625% (w/w) sodium acrylate, 2.5% (w/w) acrylamide, and 0.15% (w/w) N,N′-methylenebisacrylamide) supplemented with 0.4% (w/w) tetramethylethylenediamine (TEMED) accelerator and 0.2% (w/w) ammonium persulfate (APS) initiator. The gelation proceeded for 1 h at 37°C in a humidified incubator. Gels were further immersed into 2 ml of 8 units/ml proteinase-K in digestion buffer (pH 8.0, 50 mM Tris, 1 mM EDTA, 0.5% Triton X-100, 0.08 M guanidine HCl) solution for 4 h at 37°C for digestion. Gels were transferred to 50 ml deionized water for overnight expansion, and water was refreshed once to ensure the expansion reached plateau. Plasma-cleaned #1.5 coverslips for gel imaging were incubated in 0.1% poly-L-lysine to reduce gel's drift during acquisition. Gels were mounted using custom-printed imaging

chambers (https://www.tinkercad.com/things/7qqYCygcbNU). Expansion factor was calculated as a ratio of a gel's diameter to the diameter of gelation chamber and was in the range of 4.0–4.1.

## Confocal microscopy

Confocal microscopy of fixed samples on glass coverslips was performed with a LSM 700 confocal laser-scanning microscope (Zeiss) equipped with a Plan-Apochromat 63x NA 1.40 oil DIC, EC Plan-Neofluar 40x NA1.30 Oil DIC, and a Plan-Apochromat 20x NA 0.8 objective. Each confocal image represents a maximum intensity projection of a z-series covering the region of interest. For fluorescence intensity measurements, settings were kept the same for all conditions. Confocal microscopy of fixed samples on expanded gels was performed with a Leica TCS SP8 STED 3X microscope using a HC PL APO 63×/1.20 W CORR CS2 water immersion objective. Images were acquired with lateral pixel size in the range of 70–80 nm and axial of 180 nm using internal HyD detector. If necessary, a drift correction of Z-stack was performed in Huygens Professional version 17.04 (Scientific Volume Imaging, The Netherlands) using cross-correlation between adjacent slices. All images were deconvolved in the same program, using the CMLE algorithm, with SNR:7 and 20 iterations. Movies of 3D reconstructions of z-stacks were performed in Blender version 2.79b (Blender Institute, Amsterdam).

## Syt uptake assay

Hippocampal neurons were pretreated with 50 μM bicuculline (Sigma, 14340) in their original NB medium for 10 min at 37°C in 5% $CO_2$. Next, neurons were directly incubated for 10 min with Syt antibodies targeting the luminal side of the synaptic vesicle protein, which were diluted (1:200) in the same original NB medium supplemented with 50 μM bicuculline at 37°C in 5% $CO_2$. Next, cells were fixed, stained with secondary antibodies, and subjected to image quantifications and analysis.

## Image quantification and analysis

### Quantifications of fluorescent Syt intensity at presynaptic boutons
Presynaptic boutons were identified by swellings along the axon using expressed RFP as fill, similar as described previously (Bamji *et al*, 2003; Leal-Ortiz *et al*, 2008; Spangler *et al*, 2013). Fluorescent intensity of internalized Syt at each bouton was measured using a circular region of interest with a fixed size of ø 1.39 μm/ø 7 pix.

### SCRN1 knockdown quantifications
SCRN1 knockdown efficiency was analyzed in cortical neurons (DIV4) co-transfected with RFP and a single SCRN1 shRNA. The average fluorescence intensity was measured of the somatic region without nucleus.

### Quantifications of SCRN1 recruitment and ER morphology
For analyzing SCRN1 recruitment to ER in COS7 cells, the number of cells showing obvious enriched SCRN1 localization at ER structures was scored. For analyzing reticular ER structures in COS7 cells, the number of cells containing less than ~ 30% detectable ER tubules in cytoplasm was scored as "non-reticular ER localization".

## Live-cell imaging

Live-cell imaging (other than electric field stimulation experiments) was conducted on an inverted microscope Nikon Eclipse Ti-E (Nikon), equipped with a Plan Apo VC 100x NA 1.40 oil objective (Nikon), Plan Apo VC 60x N.A. 1.40 oil objective (Nikon), a Plan Apo VC 40x NA 1.40 oil objective (Nikon), a Yokogawa CSU-X1-A1 spinning disk confocal unit (Roper Scientific), a Photometrics Evolve 512 EMCCD camera (Roper Scientific), and an incubation chamber (Tokai Hit) mounted on a motorized XYZ stage (Applied Scientific Instrumentation), all controlled by MetaMorph (Molecular Devices) software. Cells were imaged in their original medium. During acquisition, the objective was kept at 37°C and the imaging chamber was kept at 37°C in 5% $CO_2$.

## Relative basal $Ca^{2+}$ levels' measurements

Hippocampal neurons were transfected with $Ca^{2+}$ indicators GCaMP6f or R-GECO1, and mRFP or GFP to identify transfected neurons and presynaptic boutons. Field of views with axonal structures for all conditions were selected based on similar expression levels, while remaining blind for the expression levels of $Ca^{2+}$ indicators. Duo-color time-lapses were acquired of 21 frames with 30-s time interval, with a Z-stack of three planes with 0.5-μm interval for each frame, and cells were treated with 1–10 μM ionomycin (Santa Cruz, SC3592) prior to frame 13. Fluorescent intensities of GCaMPf or R-GECO1 at single boutons were measured for the maximum intensity projections of each frame using a fixed ROI. Fluorescent values were corrected for background fluorescence and normalized to the maximum fluorescent intensity within seven frames after ionomycin treatment ($F/F_{max}$). Relative basal $Ca^{2+}$ levels were determined by the $F_0/F_{max}$ ratio, where baseline values ($F_0$) were obtained by averaging the fluorescent intensities of the five frames prior to ionomycin treatment.

## Fluorescent recovery after photo-bleaching

Fluorescent recovery after photo-bleaching experiments were conducted on the characteristic dense ER clusters in TagRFP-ER expressing hippocampal neurons showing this phenotype, or in regular dense ER structures for control conditions, using the ILas system (Roper Scientific; Fig 4E). Fluorescence recovery of TagRFP-ER in bleached regions can be interpreted as the result from two processes: (i) diffusion of TagRFP-ER within existing ER tubules (Yalcin *et al*, 2017) and (ii) local ER remodeling within the photo-bleached region. The FRAP area size and imaging settings were kept the same for all conditions. For analysis, fluorescence intensity of the bleached region was corrected for background noise and for overall bleaching occurring during acquisition. Next, the post-bleaching fluorescent recovery values were normalized to the baseline fluorescence, which was defined by the average fluorescent intensity of five initial frames prior to onset of photo-bleaching.

## Electric field stimulation and real-time $Ca^{2+}$ dynamics

All experiments were carried out in modified Tyrode's solution (pH 7.4, 25 mM HEPES, 119 mM NaCl, 2.4 mM KCl, 2 mM $CaCl_2$, 2 mM $MgCl_2$, 30 mM glucose). Objective was prewarmed to 37°C with

objective heater (Tokai Hit). Hippocampal neurons were placed in a stimulation chamber (World Precision Instruments) and stimulated (50 Aps, 20 Hz) by electric field stimulation (platinum electrodes, 10 mm spacing, 1 ms pulses of 50 mA, alternating polarity) applied by constant current stimulus isolator (WPI A 385, World Precision Instruments) in the presence of 10 μM 6-cyano-7 nitroquinoxaline-2,3-dione and 50 μM D,L-2-amino-5-phosphonovaleric acid (CNQX/AP5; Tocris Bioscience). Imaging was performed on an inverted Nikon Eclipse TE2000 microscope equipped with mercury lamp (Nikon). Fluorescence emission was detected using a 40× oil-immersion objective [Nikon Apo, numerical aperture (NA) 1.3] and ET-GFP filter (GCaMP) or ET-mCherry (mCherry), together with a EMCCD camera (Evolve 512, Photometrics) controlled by MetaMorph 7.7 software (Molecular Devices). Images were acquired every 650 ms with exposure times between 50 and 100 ms in $1 \times 1$ binning mode. Quantitative analyses of GCaMP experiments were performed with custom macros in Igor Pro (WaveMetrics) using an automated detection algorithm as described previously (Wienisch & Klingauf, 2006).

### Statistical analysis

Statistical details are included in corresponding figure legends. *P*-values are annotated as follows: $*P < 0.05$, $**P < 0.01$, and $***P < 0.001$. Data processing and statistical analysis were conducted in Prism GraphPad (version 7.0) software.

Expanded View for this article is available online.

### Acknowledgements
We thank Dr. Mike Boxem for providing the human cDNA library; Rian Stoffelen for her contributions to Fig 1E and F; and Ginny C. Farias Galdames and Arthur P.H. de Jong for critically reading the article. This work was supported by the Netherlands Organization for Scientific Research (NWO-ALW-VICI, CCH; NWO-VIDI, MA), the Netherlands Organization for Health Research and Development (ZonMW-TOP, CCH), the European Research Council (ERC; ERC-consolidator, CCH; ERC-StG, HDM), and the Proteins@Work program of the National Roadmap Large-scale Research Facilities of the Netherlands (MA).

### Author contributions
FWL designed, conducted, and interpreted experiments, and wrote the article. YC performed biochemical experiments and cloned constructs. JTK initiated the study, cloned constructs, and performed initial experiments together with MMB. AB conducted the electric field stimulation experiments, RS performed the mass spectrometry experiments and was supported by MA, and EAK provided the expansion microscopy data. HDM gave advice throughout the project and edited the article. CCH supervised the research, coordinated the study, and edited the article.

### Conflict of interest
CCH is an employee of Genentech, Inc., a member of the Roche group. The authors declare that they have no additional conflict of interest.

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
