## [Review Process File · The EMBO Journal]

VAP-SCRN1 interaction regulates dynamic endoplasmic reticulum remodeling and presynaptic function

Feline W. Lindhout, Yujie Cao, Josta T. Kevenaar, Anna Bodzeta, Riccardo Stucchi, Maria M. Boumpoutsari, Eugene A. Katrukha, Maarten Altelaar, Harold D. MacGillavry and Casper C. Hoogenraad

Review timeline:

Submission date:	12th Dec 2018
Editorial Decision:	16th Jan 2019
Revision received:	28th May 2019
Editorial Decision:	25th Jun 2019
Revision received:	11th Jul 2019
Accepted:	17th Jul 2019

Editor: Karin Dumstrei

Transaction Report:

1st Editorial Decision

16th Jan 2019

Thank you for submitting your manuscript to The EMBO Journal. Your study has now been seen by two referees and their comments are provided below.

The referees find the topic and analysis interesting, but as you can see they also find that the study "as is" is not sufficiently developed in order to consider publication here. They both point out that there is too limited mechanistic insight provided into how VAP-1/SCRN1 affects synaptic vesicle recycling and if this is mediated via effects on ER morphology/dynamics. Referee #1 suggests to look at Ca²⁺ responses, which would be one possibility.

Should you be able to extend the findings and add more mechanistic insight then I would be able to consider a revised version. Let me know if we need to discuss anything further

REFeree REPORTS:

Referee #1:

In this work Hoogenraad and collaborators study the effects of modulating the ER membrane receptors VAPA and VAPB, or of modulating secernin-1 (SCRN1, best known as a cancer biomarker) on synaptic ER structure and on synaptic vesicle recycling. They propose a model "where SV cycling is controlled by VAP/SCRN1-dependent ER remodeling".

The data in the manuscript are convincing, and the experiments are well performed. At the same time, the topic is timely, and it is important to highlight the potential activity of the presynaptic ER in synaptic transmission.

However, I find the fashion in which the manuscript is presented and in which the data are interpreted misleading. The data demonstrate thoroughly that the knock-down of proteins such as VAPA and VAPB, or SCRN1, affects the ability of synapses to recycle vesicles. However, this

cannot be taken as a direct proof for a regulation of the synaptic vesicle cycling through these proteins. The authors make this claim repeatedly:

Abstract: "We demonstrated that the VAP-SCRN1 interaction plays an important role in maintaining ER structure and dynamics, and is engaged in regulating SV cycling"

Highlights: "VAP and VAP-associated protein SCRN1 are involved in regulating SV cycling; VAP-SCRN1 interaction controls ER structure and dynamics, and SV recycling"

First subtitle in Results: "ER proteins VAPA and VAPB are involved in regulating SV cycling"

The key experiments suggesting that these proteins do not actually regulate vesicle recycling are presented in Fig. 3 and 4, and are explained by the authors quite clearly: "SCRN1 and VAP are required for proper ER morphology" (page 9), and "SCRN1 and VAP regulate ER dynamics" (page 10).

The obvious interpretation would be that VAPA and VAPB, and possibly also SCRN1, are important for the stabilization and/or the dynamics of the ER. Their perturbation affects the calcium regulation through the ER (Fig 1, and page 7: "VAP and SCRN1 are engaged in regulating evoked presynaptic Ca²⁺ responses"). Perturbing calcium dynamics in the presynapse has been known for decades to affect synaptic transmission, so there is no surprise in the observation that ER- and calcium-perturbing conditions affect synaptic transmission and vesicle recycling.

Therefore I suggest that the authors thoroughly revise their interpretations, by removing their claim that their target proteins regulate the synaptic vesicle cycle, and by focusing on the proven facts - that these proteins are important in the ER dynamics, and that their perturbation therefore affects synaptic transmission.

Referee #2:

Comments to Authors:

In this study, Lindhout and colleagues characterize the interaction of the endoplasmic reticulum (ER) protein VAP with SCRN1, and argue that this interaction is required for normal ER morphology and presynaptic function in neurons. The authors show (1) that VAP and SCRN1 are present throughout neuronal ER including axons and nerve terminals, 2) that VAP interacts with SCRN1 through an FFAT motif, and 3) that disruption of this interaction leads to defects in SV recycling and ER structure and dynamics. While the manuscript contains a lot of data (many of high quality) and the findings are of some interest, the study opens more questions than it answers and lacks the insight expected from an EMBO J paper.

Given the focus of the manuscript on SCRN1, it is surprising that the authors say so little about this protein. As VAP has many interactors, even the rationale for a focus on SCRN1 is not clearly provided. SCRN1 is not mentioned in the introduction and its potential function (not even a mention of its domain structure) not really addressed until the Discussion. Even the Discussion does not provide sufficient information to place the findings of the paper in a hypothetical functional context.

The effects of the disruption of the VAP-SCRN1 interaction on synaptic vesicle recycling and ER morphology can be very indirect and an attempt to provide a mechanistic understanding of such defects is lacking. Are the effects on synaptic vesicle recycling mediated by changes in ER morphology? Admittedly, elucidating relations of different phenotypes is often not an easy task, but this paper does not make an effort in this direction.

Knowledge about VAP and SCRN1 is only incremental as the presence of VAP in axon and nerve terminals is not a new finding (see for example Pennetta, 2002) and SCRN1 had been already been reported to be a VAP interactor (BioGrid (Huttlin, 2017)), and predicted to have a strong FFAT motif (Murphy, Levine 2016).

The assay used to determine synaptic vesicle recycling does not discriminate between defects in exocytosis or endocytosis. This impacts the possibility of formulating some hypotheses about mechanisms

The data does not prove, as claimed, that lack of VAP results in ER discontinuities in axons. The fluorescence images shown do not allow to make this point. Narrow tubular connections may be

missed at the level of resolution shown.

Other comments

- The authors comment on an enrichment of SCRNI in neuronal tissues but they have only compared the expression within neuronal tissues (Fig1G). How does this compare to other tissues?
- Are the defects in evoked Ca²⁺ observed in Fig1N exclusively related to influx? Can defects in Ca²⁺ release from the ER be contributing to the problem, considering that the integrity of ER is affected too? And if it is influx, could it be related to SOCE which could be related to the ER integrity defect?

Minor comments

- In Fig1H the authors indicate that a Myc-VAP was used for the pull-down but in the results section they indicate it was an HA-VAP.
- A nearly identical sentence is repeated twice in the Introduction (sentence starts with "The presynaptic ER structure...")
- The FFAT binding site of VAP is not localized in its C-terminal tail, as stated in the introduction
- Several FFAT-containing proteins are cytosolic proteins that do not localize on non-ER organelles, contrary to what is stated in the introduction
- Do SCRNI and SCRNI have FFAT domains? If so discuss in view of the different properties of these proteins relative to SCRNI.

1st Revision - authors' response

28th May 2019

Please see next page.

Point-to-point response to referees - EMBOJ-2018-101345. 'VAP-SCRN1 interaction regulates dynamic endoplasmic reticulum remodeling and presynaptic function'

**** Referee #1 ****

In this work Hoogenraad and collaborators study the effects of modulating the ER membrane receptors VAPA and VAPB, or of modulating secernin-1 (SCRN1, best known as a cancer biomarker) on synaptic ER structure and on synaptic vesicle recycling. They propose a model "where SV cycling is controlled by VAP/SCRN1-dependent ER remodeling". The data in the manuscript are convincing, and the experiments are well performed. At the same time, the topic is timely, and it is important to highlight the potential activity of the presynaptic ER in synaptic transmission. However, I find the fashion in which the manuscript is presented and in which the data are interpreted misleading.

Thank you for supporting our data and acknowledging the relevance of the topic. We are grateful for your insightful comments. To address these, we have made numerous textual changes and conducted additional experiments in the revised manuscript.

1) The data demonstrate thoroughly that the knock-down of proteins such as VAPA and VAPB, or SCRN1, affects the ability of synapses to recycle vesicles. However, this cannot be taken as a direct proof for a regulation of the synaptic vesicle cycling through these proteins. The authors make this claim repeatedly: Abstract: "We demonstrated that the VAP-SCRN1 interaction plays an important role in maintaining ER structure and dynamics, and is engaged in regulating SV cycling". Highlights: "VAP and VAP-associated protein SCRN1 are involved in regulating SV cycling; VAP-SCRN1 interaction controls ER structure and dynamics, and SV recycling". First subtitle in Results: "ER proteins VAPA and VAPB are involved in regulating SV cycling". The key experiments suggesting that these proteins do not actually regulate vesicle recycling are presented in Fig. 3 and 4, and are explained by the authors quite clearly: "SCRN1 and VAP are required for proper ER morphology" (page 9), and "SCRN1 and VAP regulate ER dynamics" (page 10).

Thank you for pointing this out. We agree that the effects of VAP and SCRN1 depletion on synaptic vesicle recycling are presumably indirect, most likely caused by disruptions in ER structure and dynamics. This was also pointed out in the discussion (p13): "In this study, we reported that depletion of VAP or SCRN1 results in reduced presynaptic Ca^{2+} influx and SV recycling. These data suggest that the VAP-SCRN1 interaction, which we identified as regulator of ER remodeling, plays a regulatory role in SV recycling". Hence we did not intend to claim that VAP and SCRN1 are direct regulators of synaptic vesicle cycling, and it is important that the reader does not interpret it at such. Hence we made textual changes to the revised manuscript to avoid potential misinterpretation, and removed any suggestion that could imply that VAP and SCRN1 are direct regulators of synaptic vesicle cycling.

2) The obvious interpretation would be that VAPA and VAPB, and possibly also SCRN1, are important for the stabilization and/or the dynamics of the ER. Their perturbation affects the calcium regulation through the ER (Fig 1, and page 7: "VAP and SCRN1 are engaged in regulating evoked presynaptic Ca^{2+} responses"). Perturbing calcium dynamics in the presynapse has been known for decades to affect synaptic transmission, so there is no surprise in the observation that ER- and calcium-perturbing conditions affect synaptic transmission and vesicle recycling.

The referee raises an interesting point. Indeed, it is plausible that the observed impaired ER structure/dynamics results in perturbed Ca^{2+} dynamics, which could lead to the reported effects on evoked Ca^{2+} responses. This is in line with our suggested model described in the discussion (p.14): "Possibly, decreased VAP-SCRN1 interactions that result in impaired ER structures could abolish membrane contact sites, and thereby affect presynaptic Ca^{2+} dynamics and SV recycling". We conducted novel experiments to gain more insight in the effects of VAP/SCRN1 on Ca^{2+} homeostasis and ER integrity. Indeed, we observed increased basal Ca^{2+} levels with VAP depletion or dominant-negative SCRN1-F402A expression (Fig. 5G-L). This is consistent with our previous finding on decreased evoked Ca^{2+} influx with VAP and SCRN1 depletion, as the extracellular-cytoplasmic Ca^{2+}

concentration gradient is lower (Fig. 5D-F). Additionally, to further investigate the role of SCRNI and VAP, we visualized ER nanostructures in neurons using the recently developed expansion microscopy technique. This method allows for physical magnification of the sample and has been showed to work robustly for different subcellular structures (reviewed in Wassie et al., 2019). In line with our previous findings, we observed that VAP or SCRNI depletion leads to severe discontinuity of ER structures in axons and overall less dense and irregular ER structures in different subcellular compartments (Fig. 4E; Fig EV4H,I).

3) Therefore I suggest that the authors thoroughly revise their interpretations, by removing their claim that their target proteins regulate the synaptic vesicle cycle, and by focusing on the proven facts - that these proteins are important in the ER dynamics, and that their perturbation therefore affects synaptic transmission.

We thank the referee for these suggestions. As described above, we added new experimental data providing further insights in the proposed mechanism of the referee, and thoroughly revised the text of the manuscript accordingly.

****** Referee #2 ******

Comments to Authors:

In this study, Lindhout and colleagues characterize the interaction of the endoplasmic reticulum (ER) protein VAP with SCRNI, and argue that this interaction is required for normal ER morphology and presynaptic function in neurons. The authors show (1) that VAP and SCRNI are present throughout neuronal ER including axons and nerve terminals, 2) that VAP interacts with SCRNI through an FFAT motif, and 3) that disruption of this interaction leads to defects in SV recycling and ER structure and dynamics. While the manuscript contains a lot of data (many of high quality) and the findings are of some interest, the study opens more questions than it answers and lacks the insight expected from an EMBO J paper.

Thank you for appreciating the quality of our data and your interest in our findings. We agree with the reviewer that our work still leaves some open questions. The role of generic ER function at presynaptic sites remains remarkably poorly understood, especially considering the many studies focusing on unravelling presynaptic function. This is to a large extent due to lack of tools to study axonal ER structures. We are now starting to overcome these technical limitations. Hence, only recently the relevance of the ER in controlling presynaptic function has begun to emerge, and additional studies are required to gain an advanced understanding. To this extend, our report provides one of the first mechanistic insights in the role of ER structure/dynamics on presynaptic function. The referee provides insightful comments and suggestions on our manuscript, which we highly appreciate. We added several experiments to gain more molecular insights in to the function of ER structure/dynamics at presynaptic sites, and applied textual changes where necessary.

Given the focus of the manuscript on SCRNI, it is surprising that the authors say so little about this protein. As VAP has many interactors, even the rationale for a focus on SCRNI is not clearly provided. SCRNI is not mentioned in the introduction and its potential function (not even a mention of its domain structure) not really addressed until the Discussion. Even the Discussion does not provide sufficient information to place the findings of the paper in a hypothetical functional context.

We thank the referee for bringing up these points. We did report the following findings giving insights in the molecular function of SCRNI in the Results section:

- 1. SCRNI interacts with VAP via a single FFAT-like motif (Fig. 3, Fig. EV3)*
- 2. SCRNI undergoes competitive binding for available VAP binding pockets (Fig. 2)*
- 3. SCRNI oligomerizes (Fig. EV2D,E)*
- 4. SCRNI-F402A acts like a dominant-negative (Fig. 4; Fig. EV2D,E; p. 10)*

5. *SCRN1 C69 protease domain does not exhibit proteolytic activity unlike family members SCR2 and SCR3 (Fig. EV4A-C).*

In the Discussion section, we proposed different functional models on how these findings might explain the SCR1-mediated phenotypes on impaired ER and SV cycling (p14-15). Nevertheless, we agree with the reviewer that we presented little background on VAP and SCR1 proteins and the domain structure in the initial manuscript, as we focused mainly on the cellular functions here. We have now extended the introduction of the revised manuscript with information on VAP proteins and their interactions, as well as SCR1 and its domain structures.

The effects of the disruption of the VAP-SCR1 interaction on synaptic vesicle recycling and ER morphology can be very indirect and an attempt to provide a mechanistic understanding of such defects is lacking. Are the effects on synaptic vesicle recycling mediated by changes in ER morphology? Admittedly, elucidating relations of different phenotypes is often not an easy task, but this paper does not make an effort in this direction.

The reviewer raises interesting points. We agree that it is challenging to proof causal relationships between the phenotypes, and therefore we prefer to remain cautious in making strong claims. We hypothesize that the impaired synaptic vesicle cycling can be mediated by ER perturbations. A possible mechanistic link between these two phenotypes may come from defects in ER-mediated Ca^{2+} homeostasis. To test this, we conducted new experiments to explore the role of VAP-SCR1 interactions on maintaining basal Ca^{2+} levels. Indeed, we observed that presynaptic cytoplasmic Ca^{2+} levels were elevated upon VAP depletion or dominant-negative SCR1-F402A expression (Fig. 5G-L). Together, these data points toward a model where the ER defects observed with loss of VAP-SCR1 interactions could interfere with ER-mediated Ca^{2+} homeostasis, and thereby modulate SV cycling.

Knowledge about VAP and SCR1 is only incremental as the presence of VAP in axon and nerve terminals is not a new finding (see for example Pennetta, 2002) and SCR1 had been already been reported to be a VAP interactor (BioGrid (Huttlin, 2017)), and predicted to have a strong FFAT motif (Murphy, Levine 2016).

We thank the referee for sharing the study regarding VAP localization at nerve terminals of Drosophila neurons (Pennetta et al., 2002). We added this reference to the revised manuscript. However, we do not agree with the reviewer that our findings are incremental. SCR1 was previously predicted to interact with VAP and four potential FFAT binding sites were identified. Here we showed that only one out of the four predicted FFAT-like motifs was responsible for the VAP-SCR1 interaction, and based on this information we designed tools to interfere with this interaction. As such, we could provide new insights in the role of the VAP-SCR1 interaction (in addition to SCR1 and VAP solely) on ER remodeling and presynaptic function, which are all novel findings.

The assay used to determine synaptic vesicle recycling does not discriminate between defects in exocytosis or endocytosis. This impacts the possibility of formulating some hypotheses about mechanisms.

The referee brings up a good point. We have put a lot of time and energy in trying to dissect the effects of synaptic vesicle endo- and exocytosis on VAP and SCR1 knockdown neurons using the synaptophysin-pHluorin (sypHy) assay. In brief, neurons expressing sypHy were electrically stimulated (50 APs, 20 Hz) to induce SV exo- and compensatory endocytosis, which is measured by a respectively increase and decrease of sypHy fluorescence. Next, neurons were flushed with NH_4Cl to obtain maximum fluorescent sypHy values to enable normalization of the data. In VAP knockdown neurons we observed a significant slower decay of the sypHy fluorescence, whereas the sypHy amplitude was not significantly altered (see figure below). These data imply that VAP could be more engaged in modulating SV endocytosis than SV exocytosis. However, the success rate of responses upon electric stimulation was markedly lower (~40%) in VAP knockdown neurons compared to control, which could result in a biased effect on the sypHy amplitude results. Therefore we prefer to remain cautious when interpreting this data, as we believe that more in-depth investigation is required to ultimately make a

conclusion about the specific role of VAP on endo- and exocytosis. On the other hand, in *SCRN1* knockdown neurons we unexpectedly observed a significant increased amplitude of *sypHy* fluorescence upon stimulation, which is an effect that is not commonly observed (see figure below). Thus albeit we consider this as an interesting result, it is likely that this *SCRN1*-mediated effect on SV exocytosis may occur independently from its interaction with VAP. Hence we believe that this finding is beyond the scope of the paper, and we therefore prefer to further investigate this in an independent study. We have included all the *sypHy* data for the referee's interest.

A. Average fluorescence transients of *Syp-pHluorin* in neurons co-expressing empty vector, VAPA/B shRNAs and *SCRNI* shRNA in response to 50APs at 20 Hz. Traces were normalized to the fluorescence signal during NH_4Cl superfusion at the end of the recording ($n = 26$ cells for ctrl, 13 for VAP shRNAs, 14 for *SCRNI* shRNA; from $N = 5$ independent experiments).

B. Quantification of amplitudes from A. * $p = 0.0153$

C. Average fluorescence transients of *Syp-pHluorin* in neurons co-expressing empty vector, VAP shRNAs and *SCRNI* shRNA in response to 50APs at 20 Hz ($n = 26$ cells for ctrl, 12 for VAPA/B shRNAs, 13 for *SCRNI* shRNA; from $N = 5$ independent experiments).

D. Quantification of half time ($t_{1/2}$) of recovery from C. * $p = 0.0169$

Data information: Data represent Mean \pm SEM; NS: not significant; * $P < 0.05$; ** $P < 0.01$; *** $P < 0.001$, by unpaired t -test; Box plot: Boxes represent 25th/75th percentiles, solid lines represent median, circles represents single data points, and whiskers represent minima/maxima

The data does not prove, as claimed, that lack of VAP results in ER discontinuities in axons. The fluorescence images shown do not allow to make this point. Narrow tubular connections may be missed at the level of resolution shown.

The referee refers to the fluorescent images which provided a first indication of possible ER discontinuity in the tested conditions (Fig. 4B). The actual validation for this observation comes from the FRAP analysis with an ER marker (Fig. 4F-J), which is used by us and other as quantifiably read-out for ER discontinuity (Yalcin et al., 2017). We applied textual changes to better clarify this. Nevertheless, we agree with the referee that it is not feasible to distinguish narrow ER tubules in neurons at the resolution shown (Fig. 4B). We added new experiments to visualize ER nanostructures in neurons, by increasing the resolution using the recently developed expansion microscopy technique (Fig 4C; Fig EV4E,F). This method enables physical sample enlargement and has already successfully been applied in different biomaterials to increase the resolution of various subcellular structures (reviewed in Wassie et al., 2019). Consistent with our FRAP results, we observed severely disrupted and discontinuous ER structures in axons and overall less dense and irregular ER structures in different subcellular compartments with VAP or SCRNI depletion.

Other comments

- The authors comment on an enrichment of SCRNI in neuronal tissues but they have only compared the expression within neuronal tissues (Fig1G). How does this compare to other tissues?

We thank the referee for raising this point. The enriched expression of SCRNI in brain tissue was indicated by various online protein expression databases, we now conducted more experiments to biochemically confirm these findings. We found that endogenous SCRNI expression is enriched in neuronal tissues compared to non-neuronal tissues using Western Blot analysis and two different SCRNI antibodies (Fig 1E; Fig EV1E).

- Are the defects in evoked Ca²⁺ observed in Fig1N exclusively related to influx? Can defects in Ca²⁺ release from the ER be contributing to the problem, considering that the integrity of ER is affected too? And if it is influx, could it be related to SOCE which could be related to the ER integrity defect?

The referee raises an interesting discussion point. Indeed, it is plausible that the observed phenotype on evoked Ca²⁺ influx is caused by impaired ER integrity, as this could interfere with Ca²⁺ homeostasis. This was tested by the same experiment as described before, were we observed increased cytoplasmic Ca²⁺ levels at boutons with VAP depletion or dominant-negative SCRNI-F402A expression (Fig. 5G-L). This implies that the extracellular-cytoplasmic Ca²⁺ concentration gradient is decreased, which could explain the decreased evoked Ca²⁺ influx with VAP and SCRNI depletion (Fig. 5D-F).

Minor comments

- In Fig1H the authors indicate that a Myc-VAP was used for the pull-down but in the results section they indicate it was an HA-VAP.

We thank the referee for catching this discrepancy.

- A nearly identical sentence is repeated twice in the Introduction (sentence starts with "The presynaptic ER structure...")

We thank the referee for noticing, we applied textual changes accordingly.

- The FFAT binding site of VAP is not localized in its C-terminal tail, as stated in the introduction

We thank the referee for catching this inaccuracy.

- Several FFAT-containing proteins are cytosolic proteins that do not localize on non-ER organelles, contrary to what is stated in the introduction

Thank you for bringing this up, we clarified this in the revised manuscript.

- Do SCR2 and SCR3 have FFAT domains? If so discuss in view of the different properties of these proteins relative to SCR1.

The FFAT-like motif in SCR1 is not shared within SCR2 and SCR3. However, SCR2 and SCR3 are predicted to possibly contain FFAT-like motifs, but the prediction score is low (Murphy and Levine, 2016). Thus, in our view we cannot yet be certain that SCR2 and SCR3 do or do not have functional FFAT-like motifs.

2nd Editorial Decision

25th Jun 2019

Thank you for submitting your revised manuscript. Your study has now been seen by the two referees and their comments are provided below.

Both referees appreciate that the analysis is well done and referee #1 supports publication as is. Referee #2 is hesitant if we gain enough mechanistic insight for consideration here. I see the points raised by the referee, but on balance I also find the study interesting and that it adds insight into a new function of VAP-SCRN1 in presynaptic function. I am therefore OK to move forward with the manuscript as is. However, I would like to ask you discuss the point regarding the SCRN1 kd/ dom negative constructs as brought up by referee #2. Would you also make sure that you provide a balanced discussion of what you can concluded and make sure not to overstate the findings.

REFeree REPORTS:

Referee #1:

The authors have responded to all of my comments well, and I am happy to suggest that the manuscript should be published.

Referee #2:

The authors have addressed some of my concerns. While the technical quality of the manuscript is generally high, the study fails to provide a mechanistic understanding of the stated conclusions, which I find inflated. The proposed role of SCRN1 in ER architecture remains questionable (results with SCRN1 KD data are less impressive than results with the expression of dominant negative fragments). One issue not discussed is the ER localization of some of the FFAT motif mutants, but restricted to the central E

R domains (see Fig S3A). The graphical abstract sells a highly speculative hypothesis as a fact. If this paper has to be published in the EMBO J the conclusions needs to be tempered.

2nd Revision - authors' response

11th Jul 2019

The authors performed the requested editorial changes.

3rd Editorial Decision

17th Jul 2019

Thank you for sending me the revised manuscript. I have now had a chance to take a look at it and all looks good. I am therefore very pleased to accept the manuscript for publication here.

Corresponding Author Name: Casper C. Hoogenraad

Journal Submitted to: EMBO J

Manuscript Number: EMBOJ-2018-101345